# On the Closed-Form of Flow Matching: Generalization Does Not Arise from Target Stochasticity

**Quentin Bertrand[1]**[*], **Anne Gagneux[2]**[*], **Mathurin Massias[3]**[*], **Rémi Emonet[14]**[*]

[1]Université Jean Monnet Saint-Étienne, CNRS, Institut d'Optique Graduate School,
Inria, Laboratoire Hubert Curien UMR 5516, F-42023 Saint-Étienne, France
[2]ENS de Lyon, CNRS, Université Claude Bernard Lyon 1, Inria,
LIP UMR 5668, 69342 Lyon Cedex 07, France
[3]Inria, ENS de Lyon, CNRS, Université Claude Bernard Lyon 1,
LIP UMR 5668, 69342 Lyon Cedex 07, France
[4]Institut Universitaire de France
Code: https://github.com/generativemodels/closedformfm

## Abstract

Modern deep generative models can now produce high-quality synthetic samples that are often indistinguishable from real training data. A growing body of research aims to understand why recent methods, such as diffusion and flow matching techniques, generalize so effectively. Among the proposed explanations are the inductive biases of deep learning architectures and the stochastic nature of the conditional flow matching loss. In this work, we rule out the noisy nature of the loss as a key factor driving generalization in flow matching. First, we empirically show that in high-dimensional settings, the stochastic and closed-form versions of the flow matching loss yield nearly equivalent losses. Then, using state-of-the-art flow matching models on standard image datasets, we demonstrate that both variants achieve comparable statistical performance, with the surprising observation that using the closed-form can even improve performance.

## 1 Introduction

Recent deep generative models, such as diffusion (Sohl-Dickstein et al., 2015; Ho et al., 2020; Song et al., 2021) and flow matching models (Lipman et al., 2023; Albergo and Vanden-Eijnden, 2023; Liu et al., 2023), have achieved remarkable success in synthesizing realistic data across a wide range of domains. State-of-the-art diffusion and flow matching methods are now capable of producing multi-modal outputs that are virtually indistinguishable from human-generated content, including images (Stability AI, 2023), audio (Borsos et al., 2023), video (Villegas et al., 2022; Brooks et al., 2024), and text (Gong et al., 2023; Xu et al., 2025).

A central question in deep generative modeling concerns the generalization capabilities and underlying mechanisms of these models. Generative models generalization remains a puzzling phenomenon, raising a number of challenging and unresolved questions: whether generative models truly generalize is still the subject of active debate. On one hand, several studies (Carlini et al., 2023; Somepalli et al., 2023b,a; Dar et al., 2023) have shown that large diffusion models are capable of memorizing individual samples from the training set, including licensed photographs, trademarked logos, and sensitive medical data.

On the other hand, Kadkhodaie et al. (2024) have empirically demonstrated that while memorization can occur in low-data regimes, diffusion models trained on a *sufficiently large* dataset exhibit clear

---

[*]Equal contribution. Correspondence: quentin.bertrand@inria.fr.

39th Conference on Neural Information Processing Systems (NeurIPS 2025).

signs of generalization. Taken together, recent work points to a sharp phase transition between memorization and generalization (Yoon et al., 2023; Zhang et al., 2024). Multiple theories have been proposed to explain the puzzling generalization of diffusion and flow matching models. On the one hand, Kadkhodaie et al. (2024); Kamb and Ganguli (2025); Ross et al. (2025) suggested a geometric framework to understand the inductive bias of modern deep convolutional networks on images. On the other hand, Vastola (2025) suggested that generalization is due to the *noisy* nature of the training loss. In this work, we clearly answer the following question:

> *Does training on noisy/stochastic targets improve flow matching generalization?*
> *If not, what are the main sources of generalization?*

**Contributions**.

- We challenge the prevailing belief that generalization in flow matching stems from an inherently noisy loss (Section 3.1). This assumption, largely supported by studies in low-dimensional settings, fails to hold in realistic high-dimensional data regimes.
- Instead, we observe that generalization in flow matching emerges precisely when the limited-capacity neural network fails to approximate the *optimal closed-form velocity field* (Section 3.2).
- We identify two critical time intervals, at early and late times, where *neural networks fail to approximate the optimal velocity field* (Section 3.3). We show that generalization arises predominantly early along flow matching trajectories, aligning with the transition from the stochastic to the deterministic regime of the flow matching objective.
- Finally, on standard image datasets (CIFAR-10 and CelebA), we show that explicitly regressing against the optimal closed-form velocity field does not impair generalization and can, in some cases, enhance it (Section 4).

The manuscript is organized as follows. Section 2 reviews the fundamentals of conditional flow matching and recalls the closed-form of the "optimal" velocity field. Leveraging the closed-form expression of the flow matching velocity field, Section 3 investigates the key sources of generalization in flow matching. In Section 4, we introduce a learning algorithm based on the closed-form formula. Related work is discussed in detail in Section 5.

## 2   Recalls on conditional flow matching

Let $p_0 = \mathcal{N}(0, \mathrm{Id})$ be the source distribution[2] and $p_{\text{data}}$ the data distribution. We are given $n$ data points $x^{(1)}, \ldots, x^{(n)} \sim p_{\text{data}}$, $x^{(i)} \in \mathbb{R}^d$. The goal of flow matching is to find a velocity field $u : \mathbb{R}^d \times [0, 1] \to \mathbb{R}^d$, such that, if one solves on $[0, 1]$ the ordinary differential equation

$$\begin{cases} x(0) = x_0 \in \mathbb{R}^d \\ \dot{x}(t) = u(x(t), t) \end{cases} \tag{1}$$

then the law of $x(1)$ when $x_0 \sim p_0$ is $p_{\text{data}}$: one says that $u$ *transports* $p_0$ to $p_{\text{data}}$. For every value of $t$ between 0 and 1, the law of $x(t)$ defines a *probability path*, denoted $p(\cdot|t)$ that progressively transforms $p_0$ to $p_{\text{data}}$. If one knows the velocity field $u$, new samples can then be generated by sampling $x_0$ from $p_0$, solving the ordinary differential equation, and using $x(1)$ as the generated point.

In conditional flow matching, finding such a velocity field $u$ is achieved in the following way.

(i) First, define a conditioning variable $z$ independent of $t$, *e.g.,* $z = x_1 \sim p_{\text{data}}$,

(ii) Then, chose a conditional probability path $p(\cdot|z, t)$, *e.g.,* $p(\cdot|z = x_1, t) = \mathcal{N}(tx_1, (1 - t)^2 \mathrm{Id})$.

Through the continuity equation (Lipman et al., 2024, Sec. 3.5), the choice (ii) of the conditional probability path $p(\cdot|z, t)$ defines a conditional velocity field $u^{\text{cond}}(x, z, t)$. With the choices (i) and (ii), the conditional velocity field writes

$$u^{\text{cond}}(x, z = x_1, t) = \frac{x_1 - x}{1 - t} \ . \tag{2}$$

---

[2]the choice $p_0 = \mathcal{N}(0, \mathrm{Id})$ is made for simplicity; more generic choices are possible and the reader can refer to Lipman et al. (2024); Gagneux et al. (2025); Gao et al. (2025) for deeper introductions to flow matching.

The choice (ii) of the conditional probability paths $p(\cdot|z = x_1, t)$ fully defines a probability path $p(\cdot|t)$ (by marginalization against $z$) and thus defines an *optimal velocity field* $u^\star$ (through the continuity equation), that transports $p_0$ to $p_{\text{data}}$ (Lipman et al., 2023, Thm. 1)

$$u^\star(x, t) = \mathbb{E}_{z|x,t}\, u^{\text{cond}}(x, z, t) \ . \tag{3}$$

Hence, the optimal velocity $u^\star$ could be approximated by a neural network $u_\theta : \mathbb{R}^d \times [0, 1] \to \mathbb{R}^d$ with parameters $\theta$ by minimizing

$$\mathcal{L}_{\text{FM}}(\theta) = \mathbb{E}_{\substack{t \sim \mathcal{U}([0,1]) \\ x_t \sim p(\cdot|t)}} \|u_\theta(x_t, t) - u^\star(x_t, t)\|^2 \ . \tag{4}$$

However, $u^\star$ is usually (believed) intractable, as a remedy, Lipman et al. (2023, Thm. 2) showed that $\mathcal{L}_{\text{FM}}(\theta)$ is equal, up to a constant, to the conditional flow matching loss. With the choices (i) and (ii) made above, the conditional flow matching loss reads

$$\mathcal{L}_{\text{CFM}}(\theta) = \mathbb{E}_{\substack{x_0 \sim p_0 \\ x_1 \sim p_{\text{data}} \\ t \sim \mathcal{U}([0,1])}} \|u_\theta(x_t, t) - \underbrace{u^{\text{cond}}(x_t, z = x_1, t)}_{= \frac{x_1 - x_t}{1-t} = x_1 - x_0}\|^2, \tag{5}$$

where $x_t := (1 - t)x_0 + tx_1$. The objective $\mathcal{L}_{\text{CFM}}$ is easy to approximate, since it is easy to sample from $p_0 = \mathcal{N}(0, \text{Id})$ and $\mathcal{U}([0, 1])$; sampling from $p_{\text{data}}$ is approximated by sampling from $\hat{p}_{\text{data}} := \frac{1}{n} \sum_{i=1}^n \delta_{x^{(i)}}$. Although it seems natural, replacing $p_{\text{data}}$ by $\hat{p}_{\text{data}}$ in (5) has a very important consequence: it makes the minimizer $\hat{u}^\star$ of $\mathcal{L}_{\text{FM}}$ available in closed-form, which we recall below.

**Proposition 1** (Closed-form Formula of the Optimal Velocity). *When $p_{\text{data}}$ is replaced by $\hat{p}_{\text{data}}$, with the previous choices (i) and (ii), the optimal velocity field $\hat{u}^\star$ in (3) has a closed-form formula:*

$$\hat{u}^\star(x, t) = \sum_{i=1}^n \lambda_i(x, t) \frac{x^{(i)} - x}{1 - t} \ , \tag{6}$$

*with $\lambda(x, t) = \text{softmax}((-\frac{\|x - tx^{(j)}\|^2}{2(1-t)^2})_{j=1,...,n}) \in \mathbb{R}^n$.*

The notation $\hat{u}^\star$ emphasizes the velocity field is optimal for the *empirical* probability distribution $\hat{p}_{\text{data}}$, not the true one $p_{\text{data}}$. Since $u^{\text{cond}}(x, z = x^{(i)}, t) \propto x^{(i)} - x$, the optimal velocity field $\hat{u}^\star$ is a weighted average of the $n$ different directions $x^{(i)} - x$. Note that the closed-form formula in Equation (6) can be found in various previous works, *e.g.,* Kamb and Ganguli (2025, Eq. 3), Biroli et al. (2024), Gao and Li (2024), Li et al. (2024) or Scarvelis et al. (2025), and can be generalized to other choices of continuous distribution $p_0$ (*e.g.,* the uniform distribution, see Appendix A.1).

From Equation (6), as $t \to 1$, the velocity field $\hat{u}^\star$ diverges at any point $x$ that does not coincide with one of the training samples $x^{(i)}$, and it points in the direction of the nearest $x^{(i)}$. This creates a paradox: solving the ordinary differential equation (1) with the velocity field $\hat{u}^\star$ can only produce training samples $x^{(i)}$ (see Gao and Li 2024, Thm. 4.6 for a formal proof). Therefore, in practice, exactly minimizing the conditional flow matching loss would result in $u_\theta = \hat{u}^\star$, meaning the model memorizes the training data and fails to generalize. This naturally yields the following question:

*How can flow matching generalize if the optimal velocity field only generates training samples?*

## 3 Investigating the key sources of generalization

In this section, we investigate the key sources of flow matching generalization using the closed-form formula of its velocity field. First in Section 3.1 we challenge the claim that generalization stems from the stochastic approximation $u^{\text{cond}}$ of the optimal velocity field $\hat{u}^\star$. Then, in Section 3.2 we show that generalization arises when $u_\theta$ fails to approximate the perfect velocity $\hat{u}^\star$. Interestingly, the target velocity estimation particularly fails at two critical time intervals. Section 3.3 shows that one of these critical times is particularly important for generalization.

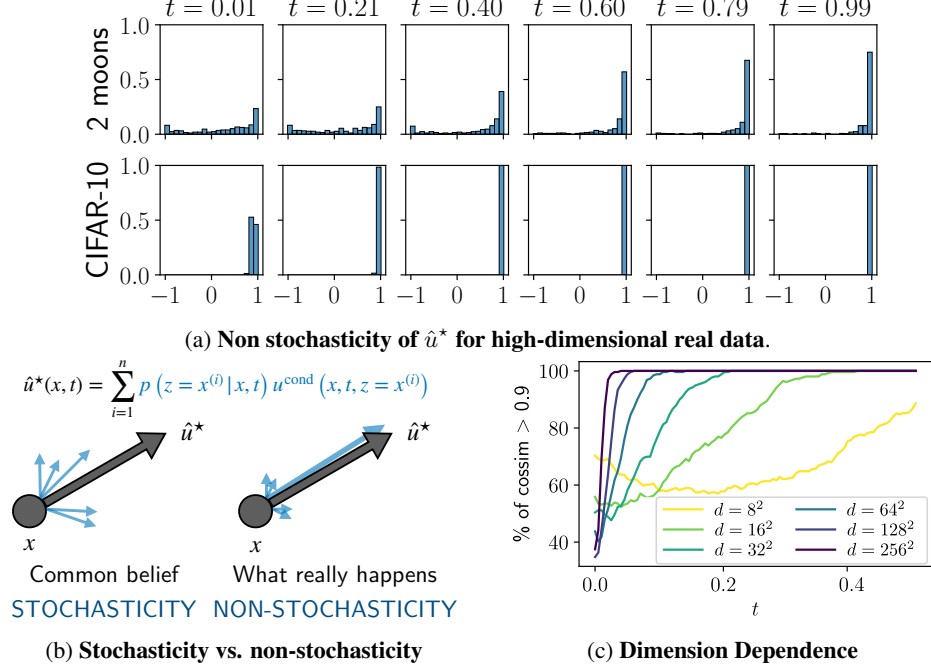

(a) **Non stochasticity of $\hat{u}^\star$ for high-dimensional real data**.

$$\hat{u}^\star(x,t) = \sum_{i=1}^{n} p\left(z = x^{(i)} | x, t\right) u^{\mathrm{cond}}\left(x, t, z = x^{(i)}\right)$$

Common belief  •  What really happens
STOCHASTICITY  NON-STOCHASTICITY

(b) **Stochasticity vs. non-stochasticity**

(c) **Dimension Dependence**

Figure 1: We challenge the hypothesis that target stochasticity plays a major role in flow matching generalization. In Figure 1a, the histograms of the cosine similarities between $\hat{u}^\star((1-t)x_0 + tx_1, t)$ and $u^{\mathrm{cond}}((1-t)x_0 + tx_1, z = x_1, t) = x_1 - x_0$ are displayed for various time values $t$ and two datasets. *For real, high-dimensional data, non-stochasticity arises very early* (before $t = 0.2$ for CIFAR-10 with dimension $(3, 32, 32)$). Figure 1c displays the alignment between $\hat{u}^\star$ and $u^{\mathrm{cond}}$ over time for varying image dimensions $d$ on Imagenette.

## 3.1 Target stochasticity is not what you need

One recent hypothesis is that generalization arises from the fact that the regression target $u^{\mathrm{cond}}$ of conditional flow matching is only a stochastic estimate of $\hat{u}^\star$. The fact that the target regression objective only equals the true objective on average is referred to by Vastola (2025) as "generalization through variance". To challenge this assumption, we leverage Proposition 1, which states that the optimal velocity field $\hat{u}^\star(x,t)$ is a weighted sum of the $n$ values of $u^{\mathrm{cond}}(x, t, z = x^{(i)}) = \frac{x^{(i)} - x}{1-t}$, for $i \in [n]$, and show that, after a *small time value* $t$, this average is in practice equal to a single value in the expectation (see Figures 1a and 1b).

**Comments on Figure 1a**. To produce Figure 1a, we sample 256 pairs $(x_0, x_1)$ from $p_0 \times \hat{p}_{\mathrm{data}}$. For each value of $t$, we compute the cosine similarity between the optimal velocity field $\hat{u}^\star((1 - t)x_0 + tx_1, t)$ and the conditional target $u^{\mathrm{cond}}((1 - t)x_0 + tx_1, z = x_1, t) = x_1 - x_0$. The resulting similarities are aggregated and shown as histograms. The top row displays the results for the two-moons toy dataset ($d = 2$), and the bottom row displays the results for the CIFAR-10 dataset (Krizhevsky and Hinton 2009, $d = 3072$); $n = 50k$ for both. As $t$ increases, the histograms become increasingly concentrated around 1, indicating that $\hat{u}^\star$ aligns closely with a single conditional vector $u^{\mathrm{cond}}$. From Equation (6), this corresponds to a collapse towards 0 of all but one of the softmax weights $\lambda_i(x_t, t)$. This time corresponds to the collapse time studied by Biroli et al. (2024) for diffusion; we discuss the connection in the related works (Section 5). On the two-moons toy dataset, this transition occurs for intermediate-to-large values of $t$, echoing the observations made in low-dimensional settings by Vastola (2025, Figure 1). In contrast, for high-dimensional real datasets, $\hat{u}^\star(x, t)$ aligns with a single conditional velocity field $x^{(i)} - x$, even at early time steps, suggesting that the non-stochastic regime dominates most of the generative process. This key difference between low- and high-dimensional data suggests that the transition time between the stochastic and non-stochastic regimes is strongly influenced by the dimensionality of the data.

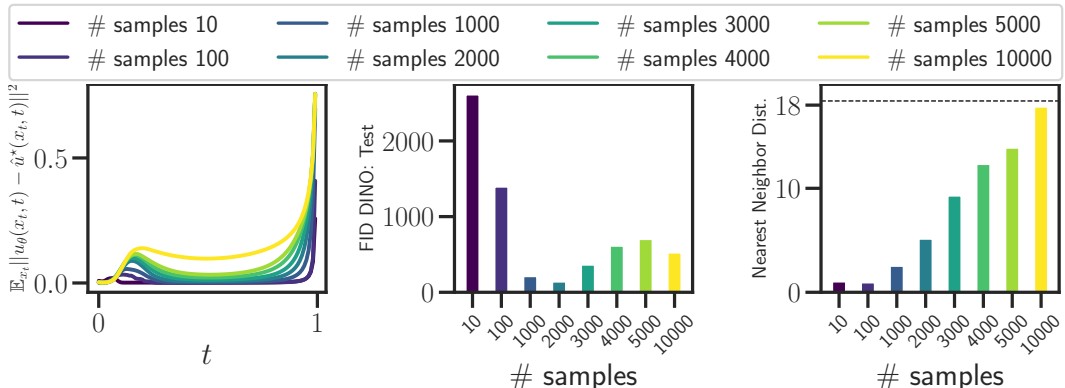

Figure 2: **Failure to learn the optimal velocity field, CIFAR-10**. *Left*: The leftmost figure represents the average error between the optimal empirical velocity field $\hat{u}^\star$ and the learned velocity $u_\theta$ for multiple values of time $t$. *Middle*: The middle figure displays the FID-10k computed on the test dataset, using the DINOv2 embedding. *Right*: The rightmost figure displays the average distance between the generated samples and their closest image from the training set – for reference, the horizontal dashed line indicates the mean distance between an image of CIFAR-10 train and its nearest neighbor in the dataset. All the quantities are computed/learned on a varying number of training samples ($10$ to $10^4$) of the CIFAR-10 dataset.

**Comments on Figure 1c**. To further illustrate the strong impact of dimensionality, Figure 1c reports the proportion of samples $x_t$ (from a batch of 256) for which the cosine similarity between $\hat{u}^\star$ and $u^{\mathrm{cond}} \propto x^{(i)} - x$ exceeds $0.9$, as a function of time $t$. This analysis is performed across multiple spatial resolutions of the Imagenette dataset (Howard, 2019), obtaining $\dim \times \dim$ images by spatial subsampling. Figure 1c reveals a sharp transition: as the dimensionality increases, the proportion of high-cosine matches rapidly converges to 100%. A practical implication of this behavior is that, for sufficiently large $t$, if $x_0 \sim p_0$ and $x^{(i)} \sim \hat{p}_{\mathrm{data}}$, then $\hat{u}^\star((1-t)x_0 + tx^{(i)}, t)$ is approximately proportional to $x^{(i)} - x$. Consequently, regressing on $x^{(i)}$ or on the conditional velocity $x_1 - x_0$ becomes effectively equivalent. Section 4 investigates how to learn regressing against optimal velocity field $\hat{u}^\star$, and empirically shows similar results between stochastic and non-stochastic targets.

The regime where flow matching matches stochasticity is mostly concentrated on a very short time interval, for small values of $t$. We hypothesize that the phenomenon observed here on the optimal velocity field $\hat{u}^\star$ has major implications on the *learned* flow matching model $u_\theta$, which we further inspect in the next section.

### 3.2 Failure to learn the optimal velocity field

This subsection investigates how well the learned velocity field $u_\theta$ approximates the optimal/ideal velocity field $\hat{u}^\star$, and how the quality of this approximation correlates with generalization. To do so, we propose the following experiment.

**Set up of Figure 2**. To build Figure 2, we subsampled the CIFAR-10 dataset from $10$ to $10^4$ samples. For each size, we trained a flow matching model using a standard 34 million-parameter U-Net (see Appendix D for details). Following Kadkhodaie et al. (2024), the number of parameters of the network $u_\theta$ remains fixed across dataset sizes. Importantly, the optimal velocity field $\hat{u}^\star$ itself depends on the dataset size: as the number of samples increases, the complexity of $\hat{u}^\star$ also grows. Thus, we expect the network $u_\theta$ to accurately approximate the optimal velocity field $\hat{u}^\star$ for smaller dataset sizes.

**Comments on Figure 2**. The leftmost plot shows the average training error

$$\mathbb{E}_{\substack{x_0 \sim p_0 \\ x_1 \sim \hat{p}_{\mathrm{data}}}} \|u_\theta(x_t, t) - \hat{u}^\star(x_t, t)\|^2, \quad \text{where} \quad x_t := (1-t)x_0 + tx_1 ,$$

between the learned velocity $u_\theta$ and the optimal empirical velocity field $\hat{u}^\star$, evaluated across multiple time values $t$ and dataset sizes. With only $10$ samples (darkest curve), the network $u_\theta$ closely approximates $\hat{u}^\star$. As the dataset size increases, the complexity of $\hat{u}^\star$ grows, and the approximation by $u_\theta$ becomes less accurate. In particular, the approximation fails at two specific time intervals: around

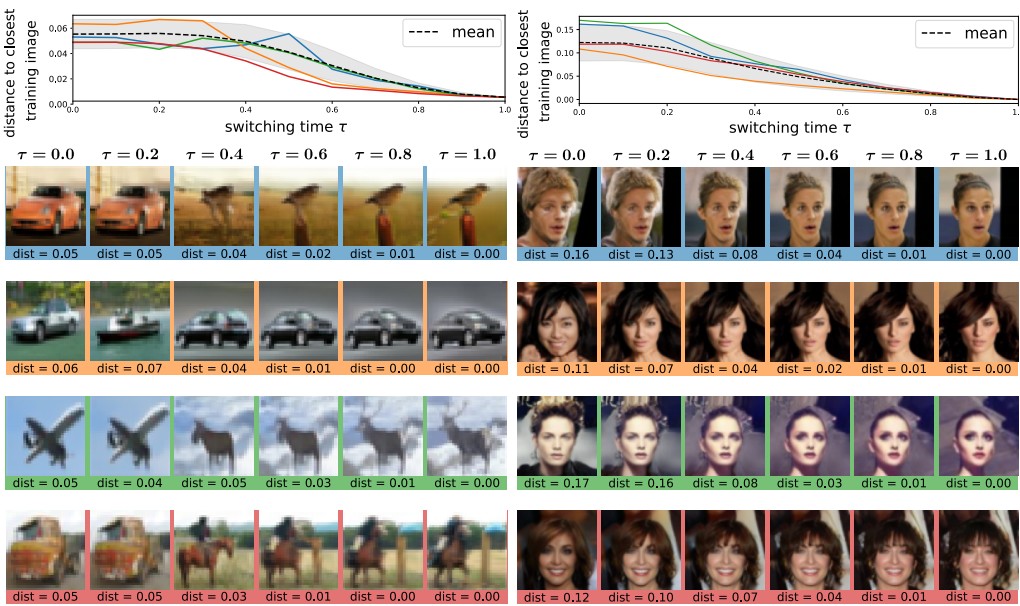

Figure 3: **Generalization occurs at small times on CIFAR-10 (left) and CelebA** 64 **(right)**. *Top*: Generalization (distance between generated samples and training data) of hybrid models that follow $\hat{u}^\star$ on $[0, \tau]$, then $u_\theta$ on $[\tau, 1]$. The four colored curves correspond to four specific $x_0$, the black dashed curve is the mean distance over the 256 generated images. *Bottom*: visualization of generated images for the four different starting noises and various values of $\tau$ (the background color matching the curve in the top figure). *Following $\hat{u}^\star$ until $\tau \geq 0.3$ yields a model that is not able to generalize.*

$t \approx 0.15$ and near $t = 1$. The failure near $t = 1$ is expected, as $\hat{u}^\star$ becomes non-Lipschitz at $t = 1$. Interestingly, the early-time failure at $t \approx 0.15$ corresponds to the regime where $\hat{u}^\star$ and $u^{\text{cond}}$ start to correlate (see Figure 1a in Section 3.1). The middle plot of Figure 2 reports the FID-10k, computed on the test set in the DINOv2 embedding space (Oquab et al., 2024), for various dataset sizes. For a small dataset (*e.g.,* #samples $= 10$), $u_\theta$ approximates $\hat{u}^\star$ well but does not generalize – the test FID exceeds $10^3$. As the dataset size increases ($1000 \leq$ #samples $\leq 3000$), the approximation $u_\theta$ becomes less accurate. Despite this, the model achieves lower FID scores on the test set but still memorizes the training data. The rightmost plot of Figure 2 illustrates this memorization by showing the average distance between each generated sample and its nearest neighbor in the training set. For larger datasets (#samples $\geq 3000$), this distance increases substantially, indicating that the model generalizes better. Overall, Figure 2 also suggests that the FID metric can be misleading, even when computed on the test set. For example, the model trained with 1000 samples has a low test FID but memorizes training examples.

Figure 2 confirms that generalization arises when the network $u_\theta$ fails to estimate the optimal velocity field $\hat{u}^\star$, and that this failure occurs at two specific time intervals. In Section 3.3, we investigate which of these two intervals is responsible for driving generalization.

## 3.3 When does generalization arise?

To investigate whether the failure to approximate $\hat{u}^\star$ matters the most at small or large values of $t$, we carry out the following experiment.

**Set up of Figure 3**. We first learn a velocity field $u_\theta$ using standard conditional flow matching (see Appendix D), then we construct a hybrid model: we define a piecewise trajectory where the flow is governed by the optimal velocity field $\hat{u}^\star$ for times $t \in [0, \tau]$, and by the learned velocity field $u_\theta$ for times $t \in [\tau, 1]$, for a given threshold parameter $\tau \in [0, 1]$. For the extreme case $\tau = 1$, the full trajectory follows $\hat{u}^\star$, and samples exactly match training data points. Conversely, when $\tau = 0$, the entire trajectory is governed by $u_\theta$, yielding novel samples. Intermediate values of $\tau$ produce a mixture of both behaviors, which we interpret as reflecting varying degrees of generalization. To assess generalization, we measure the distance of generated samples to the dataset using the LPIPS

metric (Zhang et al., 2018), which computes the feature distance between two images via some pretrained classification network. We define the distance of a generated sample $x$ to a dataset $\mathcal{D} = \{x^{(1)}, \ldots, x^{(n)}\}$ as $\mathrm{dist}(x, \mathcal{D}) = \min_{x^{(i)} \in \mathcal{D}} \mathrm{LPIPS}(x, x^{(i)})$. We fix a random batch of 256 pure noise images from $p_0$. Then, for various threshold values $\tau$, we generate 256 images with the hybrid model, always starting from this batch. Finally, we measure the creativity of the hybrid model as the mean of the aforementioned LPIPS distances between the 256 generated samples and the dataset.

**Comments on Figure 3**. The top row displays the LPIPS distances as $\tau$ varies, on the CIFAR-10 (left) and CelebA - $64 \times 64$ (right) datasets. For $\tau \leq 0.2$, the hybrid model remains as creative as $u_\theta$, despite following $\hat{u}^\star$ in the first steps. For $\tau > 0.2$, the LPIPS distance starts dropping. On the displayed generated samples (bottom rows), we in fact see that as soon as $\tau \geq 0.4$, the sample generated by the hybrid model is almost the same as the one obtained with $\hat{u}^\star$ ($\tau = 1$). This means *the final image is already determined at* $t = 0.4$, and despite the generalization capacity of the learned velocity field $u_\theta$, following it only after $t \geq 0.4$ is not enough to create a new image: *generalization occurs early and cannot fully be explained by the failure to correctly approximate* $u^\star$ *at large t.*

Although we have shown that the stochastic phase was limited to small values of $t$ in real-data settings, we have not yet definitively ruled it out as the cause of generalization. In the following Section 4, we introduce a learning procedure designed to address this question directly.

# 4 Learning with the closed-form formula

In this section, in order to discard the impact of stochastic target on the generalization, we propose to directly regress against the closed-form formula in Equation (6).

## 4.1 Empirical flow matching

Regressing against the closed-form $\hat{u}^\star$, defined in Equation (6), at a point $(x_t, t)$ requires computing a weighted sum of the conditional velocity fields over *all* the $n$ training points $x^{(i)}$. For a dataset of $n$ samples of size $d$, and a batch of size $|\mathcal{B}|$, computing the weights of the exact closed-form formula $\hat{u}^\star(x, t)$ of flow matching requires $\mathcal{O}(n \times |\mathcal{B}| \times d)$. These computations are prohibitive since they must be performed for each batch. One natural idea is to estimate the closed-form formula $\hat{u}^\star$ (Equation (6)), by a Monte Carlo approximation (Equation (8)), using $M \leq n$ samples $b^{(1)}, \ldots, b^{(M)}$:

$$\mathcal{L}_{\mathrm{EFM}}(\theta) = \mathbb{E}_{\substack{x_0 \sim p_0 \\ x_1 \sim \hat{p}_{\mathrm{data}} \\ t \sim \mathcal{U}([0,1]) \\ b^{(2)}, \ldots, b^{(M)} \sim \hat{p}_{\mathrm{data}}}} \|u_\theta(x_t, t) - \hat{u}_M^\star(x_t, t)\|^2 \ , \tag{7}$$

with $x_t = (1 - t)x_0 + tx_1$, $b^{(1)} := x_1$, and

$$\hat{u}_M^\star(x, t) = \sum_{j=1}^{M} \lambda(x, t) \frac{b^{(j)} - x}{1 - t} \ , \quad \lambda(x, t) = \mathrm{softmax}\left(\left(-\frac{\|x - tx^{(l)}\|^2}{2(1 - t)^2}\right)_{l=1,\ldots,n}\right) \ . \tag{8}$$

The formulation in Equation (7) may appear naive at first glance. Still, it hinges on a crucial trick: the Monte Carlo estimate is computed using a batch that systematically includes the point $x_1$, that generated the current $x_t$. If instead $b^{(1)}$ were sampled independently from $\hat{p}_{\mathrm{data}}$, this could introduce a sampling bias (see Ryzhakov et al. 2024, Appendix B, and the corresponding OpenReview comments[3] for an in-depth discussion). Proposition 2 shows that the estimate $\hat{u}_M^\star$ is unbiased and has lower variance than the standard conditional flow matching target.

**Proposition 2.** *We denote the conditional probability distribution $p(z = x^{(i)} \mid x, t)$ over $\{x^{(i)}\}_{i=1}^n$ by $\hat{p}_{\mathrm{data}}(z \mid x, t)$. With no constraints on the learned velocity field $u_\theta$,*

  *i) The minimizer of Equation (7) writes, for all $(x, t)$*

$$\mathbb{E}_{\substack{b^{(1)} \sim \hat{p}_{\mathrm{data}}(\cdot \mid x, t) \\ b^{(2)}, \ldots, b^{(M)} \sim \hat{p}_{\mathrm{data}}}} [\hat{u}_M^\star(x, t)] \ . \tag{9}$$

---

[3] https://openreview.net/forum?id=XYDMAckWMa

*ii) In addition, for all $(x,t)$, the minimizer of Equation (7) equals the optimal velocity field, i.e.,*

$$\mathbb{E}_{\substack{b^{(1)} \sim \hat{p}_{\text{data}}(\cdot|x,t) \\ b^{(2)},\ldots,b^{(M)} \sim \hat{p}_{\text{data}}}} [\hat{u}_M^\star(x,t)] = \hat{u}^\star(x,t) \ . \tag{10}$$

*iii) The conditional variance of the estimator $\hat{u}_M^\star$ is smaller than the usual conditional variance:*

$$\text{Var}_{\substack{b^{(1)} \sim \hat{p}_{\text{data}}(\cdot|x,t) \\ b^{(2)},\ldots,b^{(M)} \sim \hat{p}_{\text{data}}}} [\hat{u}_M^\star(x,t)] \leq \text{Var}_{b^{(1)} \sim \hat{p}_{\text{data}}(\cdot|x,t)} \left[ u^{\text{cond}}(x,b^{(1)},t) \right]. \tag{11}$$

The proof of Proposition 2 is provided in Appendix B.3. The estimator $\hat{u}_M^\star$ of the optimal field $\hat{u}^\star$ is closely related to self-normalized importance sampling (see Appendix B.2 and Owen 2013, Chap. 9.2), as well as to Rao-Blackwellized estimators (Casella and Robert, 1996; Cardoso et al., 2022). As discussed in Ryzhakov et al. (2024), self-normalized importance sampling estimators of $\hat{u}^\star$ are generally biased, in the sense that: $\mathbb{E}_{b^{(1)},\ldots,b^{(M)} \sim \hat{p}_{\text{data}}} \hat{u}_M^\star(x_t,t) \neq \hat{u}^\star(x_t,t)$ . A key insight is that our estimator includes $b^{(1)} \sim \hat{p}_{\text{data}}(\cdot \mid x_t,t)$, which leads to the main result of Proposition 2. In Section 4.2, we demonstrate that Algorithm 2, designed to solve Equation (7), yields consistent improvements on high-dimensional datasets such as CIFAR-10 and CelebA. Additional details on the unbiasedness of $\mathcal{L}_{\text{EFM}}$ can be found in the supplementary material (Appendix B). From a computational perspective, despite requiring $M$ additional samples, Algorithm 2 remains significantly more efficient than increasing the batch size by a factor of $M$: the $M$ samples are merely averaged (with weights), while the backpropagation remains identical to that of Algorithm 1.

| **Algorithm 1** Vanilla Flow Matching | **Algorithm 2** Empirical Flow Matching |
|---|---|
| **for** $k$ *in* $1,\ldots,n_{\text{iter}}$ **do** | **param :** $M$ // Number of samples in the empirical mean |
| $\quad t \sim \mathcal{U}([0,1])$ | **for** $k$ *in* $1,\ldots,n_{\text{iter}}$ **do** |
| $\quad x_0 \sim \mathcal{N}(0,\text{Id}), x_1 \sim \hat{p}_{\text{data}},$ | $\quad x_0 \sim \mathcal{N}(0,\text{Id}), x_1 \sim \hat{p}_{\text{data}}, t \sim \mathcal{U}([0,1])$ |
| $\quad x_t = (1-t)x_0 + tx_1$ | $\quad x_t = (1-t)x_0 + tx_1$ |
| $\quad u^{\text{cond}}(x_t,t) = \dfrac{x_1 - x_t}{1-t} = x_1 - x_0$ | $\quad b^{(1)} = x_1$ |
| $\quad \mathcal{L}(\theta) = \left\| u_\theta(x_t,t) - u^{\text{cond}}(x_t,t) \right\|^2$ | $\quad \forall j \in [\![2,M]\!], b^{(j)} \sim \hat{p}_{\text{data}}$ // Samples from $\hat{p}_{\text{data}}$ |
| $\quad$ Compute $\nabla\mathcal{L}(\theta)$ and update $\theta$ | $\quad \hat{u}_M^\star(x_t,t) = \sum_{j=1}^{M} \frac{b^{(j)} - x_t}{1-t} \cdot \left[ \text{softmax}\left( -\frac{\|x_t - t \cdot b\|^2}{2(1-t)^2} \right) \right]_j$ |
| **return** $u_\theta$ | $\quad \mathcal{L}(\theta) = \left\| u_\theta(x_t,t) - \hat{u}_M^\star(x_t,t) \right\|^2$ |
| | $\quad$ Compute $\nabla\mathcal{L}(\theta)$ and update $\theta$ |
| | **return** $u_\theta$ |

## 4.2 Experiments

We now learn with empirical flow matching (EFM, Equation (7) and Algorithm 2) in practical high-dimensional settings. Our goal with this empirical investigation is first to observe if regressing against a more deterministic target leads to performance improvement/degradation.

**Datasets and Models**. We perform experiments on the image datasets CIFAR-10 (Krizhevsky and Hinton, 2009) and CelebA $64 \times 64$ (Liu et al., 2015). For the experiments, we compare vanilla conditional flow matching (Lipman et al., 2023; Liu et al., 2023; Albergo and Vanden-Eijnden, 2023), optimal transport flow matching (Pooladian et al., 2023; Tong et al., 2024), and the empirical flow matching in Algorithm 2, for multiple numbers of samples $M$ to estimate the empirical mean. Training details are in Appendix D.

**Metrics**. To assess generalization performance, we use the standard Fréchet Inception Distance (Heusel et al., 2017) with Inception-V3 (Szegedy et al., 2016) but we also follow the recommendation of Stein et al. (2023) using the DINOv2 embedding (Oquab et al., 2023), which is known to a more expressive and discriminative embedding, that leads to a less biased evaluation. We also measure the FID between the generated and the train and test sets, rather than only on the training set, as is often done in generative modeling benchmarks. On Figure 2, we also displayed a memorization metric that would detect a pure copy of the training set. Overall, defining and quantifying the generalization ability of generative models is overall a challenging task: train and test FID are known to be imperfect (Stein et al., 2023; Jiralerspong et al., 2023; Parmar et al., 2022), yet no superior competitor has emerged.

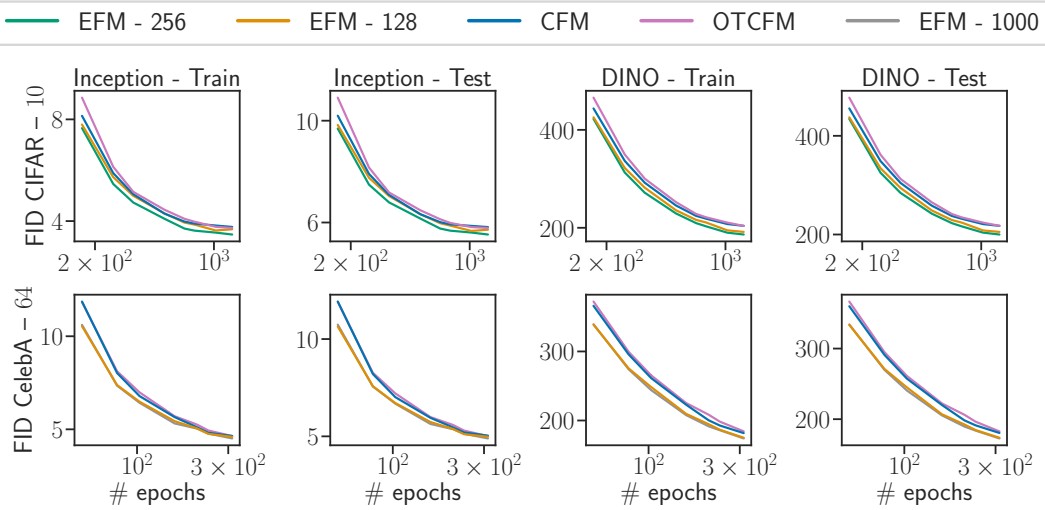

Figure 4: FID computed on the training set (50k) and the test set (10k) using multiple embeddings, Inception and DINOv2. Regressing against a more deterministic target (EFM - 128, 256, 1000) does not yield performance decreases. On the contrary, the more deterministic the target, the better the performance.

**Comments on Figure 4.** Figure 4 compares vanilla flow matching, OTCFM, and the empirical flow matching (EFM, Algorithm 2) approaches using various numbers of samples to estimate the empirical mean, $M \in \{128, 256, 1000\}$. First, we observe that learning with a more deterministic target does not degrade either training or testing performance, across both types of embeddings. On the contrary, we consistently observe modest but steady improvements as stochasticity is reduced. For both CIFAR-10 and CelebA, increasing the number of samples $M$ used to compute the empirical mean—*i.e.,* , making the targets less stochastic—leads to more stable improvements. It is worth noting that Algorithm 2 has a computational complexity of $\mathcal{O}(M \times |\mathcal{B}| \times d)$, where $|\mathcal{B}|$ is the batch size, $M$ is the number of samples used to estimate the empirical mean, and $d$ is the sample dimension. In our experiments, choosing $M = |\mathcal{B}| = 128$ yielded a modest time overhead. For empirical flow matching, we experimented with several values beyond $M = 1000$ (*e.g., M = 2000, M = 5000*). The results were nearly identical to those obtained with $M = 1000$, with curves being visually indistinguishable. Therefore, we chose not to report results for $M \geq 1000$.

## 5 Related work

The existing literature related to our study can be roughly divided into three approaches: leveraging the closed-form, studies on the memorization vs generalization, and characterization of the different phases of the generating dynamics.

**Leveraging the closed-form**. Proposition 1 has been leveraged in several ways. The closest existing work is by Ryzhakov et al. (2024), who propose to regress against $\hat{u}^\star$ as we do in Section 4. Nevertheless, their motivation is that reducing the variance of the velocity field estimation makes learning more accurate: as explained in Section 3.1, we argue this claim rests on misleading 2D-based intuitions (*e.g.,* Figure 1, challenged by Section 3.1). The idea of regressing against a more deterministic target (as Proposition 2 shows) derived from the optimal closed-form velocity field has also been empirically explored for diffusion models (Xu et al., 2023). Scarvelis et al. (2025) bypass training, and suggest using a smoothed version of $\hat{u}^\star$ to generate novel samples. In a work specific to images and convolutional neural networks, Kamb and Ganguli (2025) suggested that flow matching indeed ends up learning an optimal velocity, but that instead of memorizing training samples, the velocity memorizes a combination of all possible patches in an image and across the images. They show remarkable agreement between their theory and the trajectories followed by learned vector fields, but their work is limited to convolutional architectures, and was recently extended to a larger class of architectures (Lukoianov et al., 2025).

**Memorization and reasons for generalization**. Kadkhodaie et al. (2024) directly relates the transition from memorization to generalization to the size of the training dataset, and proposes a geometric interpretation. We provide a complementary experiment in Section 3.2, quantifying how much the network fails to estimate the optimal velocity field. Gu et al. (2025) provide a detailed experimental investigation into the potential causes for generalization, primarily based on the characteristics of the dataset and choices for training and model. Vastola (2025) explores different factors of generalization in the case of diffusion, with a special focus on the stochasticity of the target objective in the learning problem. Through a physic-based modeling of the generative dynamics, they study the covariance matrix of the noisy estimation of the exact score. In our work, we believe that we have shown that this claim was not valid for real high-dimensional data. Niedoba et al. (2025) study the poor approximation of the exact score by the learned models: like Kamb and Ganguli (2025), they suggest that the generalization of the learned models comes from memorization of many patches in the training data.

**Temporal regimes**. Biroli et al. (2024); Sclocchi et al. (2025) provide an analysis of the exact score, the counterpart of the exact velocity field for diffusion. For a multimodal target distribution, the authors identify three phases (we keep the convention that $t = 0$ is noise and $t = 1$ is target): for $t < t_1$, all trajectories are indistinguishable; for $t_1 < t < t_2$, trajectories converging to different modes separate; for $t > t_2$, trajectories all point to the training dataset. In the case of Gaussians mixtures target, they highlight the dependency of $t_2$ in the dimension and the number of samples, in $\mathcal{O}\left((\log n)/d\right)$, meaning that the first phases are observable only if the number of training points is exponential in the dimension. The methodology they adopt to validate the existence of such $t_2$ on real data relies on the stochasticity of the backward generative process, which does not hold in the case of flow matching. Our experiments on *learned* flow matching models allow us to take this theoretical study on memorization and temporal behaviors of generative processes a step further.

# 6    Conclusion, limitations and broader impact

**Conclusion**. By challenging the assumption that stochasticity in the loss function is a key driver of generalization, our findings help clarify the role of approximation of the exact velocity field in flow matching models. Beyond the different temporal phases in the generation process that we have identified, we expect further results to be obtained by uncovering new properties of the true velocity field.

**Limitation**. Our work is mainly empirical, with a focus on *learned* models, but did not precisely characterize the learned velocity field, in particular, how it behaves outside the trajectories defined by the optimal velocity. Leveraging existing work on the inductive biases of the architectures at hand seems like a promising venue. Another limitation is that we did not investigate the interaction between the architectural inductive bias, and optimization procedures: this is a very challenging, but active area of research (Boursier and Flammarion, 2025; Bonnaire et al., 2025; Favero et al., 2025).

**Broader impact**. We hope that identifying the key factors of generalization will lead to improved training efficiency. However, generative models also raise concerns related to misinformation (notably deepfakes), data privacy, and potential misuse in generating synthetic but realistic content.

# 7 Acknowledgments

The authors thank the Blaise Pascal Center for its computational support, using the SIDUS (Quemener and Corvellec, 2013) solution.

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

# A  Proofs of Section 2

$$\hat{u}^{\star}(x,t) = \sum_{i=1}^{n} u^{\mathrm{cond}}(x, z = x^{(i)}, t) \cdot \frac{p(x|z = x^{(i)}, t)}{\sum_{i'=1}^{n} p(x|z = x^{(i')}, t)} \quad . \tag{12}$$

## A.1  Proof of Proposition 1

*Proof.* • In the case where $z \sim \hat{p}_{\mathrm{data}}$, conditional probability writes

$$p(z = x^{(i)}|x,t) = \frac{p(x, t, z = x^{(i)})}{p(x, t)} \tag{13}$$

$$= \frac{p(x|t, z = x^{(i)})p(t, z = x^{(i)})}{p(x, t)} \tag{14}$$

$$= \frac{p(x|t, z = x^{(i)})p(t, z = x^{(i)})}{\sum_{i'=1}^{n} p(x, t, z = x^{(i')})} \tag{15}$$

$$= \frac{p(x|t, z = x^{(i)})p(t)\overbrace{p(z = x^{(i)})}^{\frac{1}{n}}}{\sum_{i'=1}^{n} p(x|t, z = x^{(i')})p(t)\underbrace{p(z = x^{(i')})}_{\frac{1}{n}}} \tag{16}$$

$$= \frac{p(x|t, z = x^{(i)})}{\sum_{i'=1}^{n} p(x|t, z = x^{(i')})} \quad . \tag{17}$$

Pluging Equation (17) in Equation (3) yields the closed-formed formula for the velocity field:

$$u^{\star}(x,t) = \sum_{i=1}^{n} u^{\mathrm{cond}}(x, t, z = x^{(i)})p(z = x^{(i)}|x,t) \tag{18}$$

$$= \sum_{i=1}^{n} u^{\mathrm{cond}}(x, t, z = x^{(i)})\frac{p(x|t, z = x^{(i)})}{\sum_{i'=1}^{n} p(x|t, z = x^{(i')})} \quad . \tag{19}$$

which proves Equation (12); using that $x|t, z = x^{(i)} \sim \mathcal{N}(tx^{(i)}, (1-t)^2\,\mathrm{Id})$ and $u^{\mathrm{cond}}(x, t, z = x^{(i)}) = \frac{x^{(i)} - x}{1-t}$ yields Equation (6).

• For the case $z \sim p_0 \times \hat{p}_{\mathrm{data}}$,

$$\hat{u}^{\star}(x,t) := \int_{z} u^{\mathrm{cond}}(x, t, z)p(z|x,t)\,\mathrm{d}z \tag{20}$$

$$= \int_{z} u^{\mathrm{cond}}(x, t, z)\frac{p(x, z, t)}{p(x, t)}\,\mathrm{d}z \tag{21}$$

$$= \int_{z} u^{\mathrm{cond}}(x, t, z)\frac{p(x|z, t)p(z)p(t)}{\int_{z'} p(x|t, z')p(t)p(z')\,\mathrm{d}z'}\,\mathrm{d}z \tag{22}$$

$$= \int_{z} u^{\mathrm{cond}}(x, t, z)\frac{p(x|z, t)p(z)}{\int_{z'} p(x|t, z')p(z')\,\mathrm{d}z'}\,\mathrm{d}z \tag{23}$$

Since $z \sim p_0 \times \hat{p}_{\mathrm{data}}$, the denominator is equal to:

$$\int_{z'} p(x|t, z')p(z')\,\mathrm{d}z' = \frac{1}{n}\int_{x_0} \sum_{i=1}^{n} \delta_x((1-t)x_0 + tx^{(i)})\frac{1}{\sqrt{(2\pi)^d}}\exp(-\frac{1}{2}x_0^2)\mathrm{d}x_0 \tag{24}$$

$$= \frac{1}{n}\int_{y} \sum_{i=1}^{n} \delta_x(y)\frac{1}{\sqrt{(2\pi)^d}}\exp(-\frac{1}{2(1-t)^2}\|y - tx^{(i)}\|^2)\frac{1}{(1-t)^d}\mathrm{d}y \quad (y = (1-t)x_0 + tx^{(i)}) \tag{25}$$

$$= \frac{1}{n}\sum_{i=1}^{n} \frac{1}{\sqrt{(2\pi(1-t)^2)^d}}\exp\left(-\frac{1}{2(1-t)^2}\|x - tx^{(i)}\|^2\right) \tag{26}$$

Likewise, the numerator equals:

$$\int_z u^{\text{cond}}(x,t,z)p(x|z,t)p(z)\,\mathrm{d}z = \int_{x_0} \frac{1}{n}\sum_{i=1}^n (x^{(i)} - x_0)\delta_x((1-t)x_0 + tx^{(i)})\frac{1}{\sqrt{(2\pi)^d}}\exp(-\frac{1}{2}\|x_0\|^2)\mathrm{d}x_0$$

(27)

$$= \frac{1}{n}\sum_{i=1}^n \int_y \frac{x^{(i)} - y}{1-t}\delta_x(y)\frac{1}{\sqrt{(2\pi(1-t)^2)^d}}\exp\left(-\frac{1}{2(1-t)^2}\|y - tx^{(i)}\|^2\right)\mathrm{d}y$$

(28)

$$= \sum_{i=1}^n \frac{x^{(i)} - x}{1-t}\frac{1}{\sqrt{(2\pi(1-t)^2)^d}}\exp\left(-\frac{1}{2(1-t)^2}\|x - tx^{(i)})\|^2\right)$$

(29)

Taking the ratio of Equations (24) and (29) concludes the proof. $\qquad\square$

## B  Additional details and comments on empirical flow matching

First, recalls on the optimal velocity (Equation (6)) and the empirical flow matching loss (Equations (7) and (8)) are provided in Appendix B.1. The unbiasedness of the estimator is presented in Appendix B.2, and its proof is in Appendix B.3.

### B.1  Recalls

The closed-form formula of the "optimal" velocity field is:

$$\hat{u}^\star(x,t) = \sum_{l=1}^n \frac{x^{(l)} - x}{1-t} \cdot \left[\text{softmax}\left(\left(-\frac{\|x - tx^{(k)}\|^2}{2(1-t)^2}\right)_{k=1,\ldots,n}\right)\right]_l .$$

(6)

The proposed loss uses mini-batches of size $M$ (instead of all $n$ training points) to build an estimator $\hat{u}_M^\star$ of $\hat{u}^\star$:

$$\mathcal{L}_{\text{EFM}}(\theta) = \mathbb{E}_{\substack{t\sim\mathcal{U}([0,1])\\ x_0\sim p_0\\ x_1\sim\hat{p}_{\text{data}}\\ x_t=(1-t)x_0+tx_1\\ b^{(1)}:=x_1 \,;\, b^{(2)},\ldots,b^{(M)}\sim\hat{p}_{\text{data}}}} \|u_\theta(x_t,t) - \hat{u}_M^\star(x_t,t)\|^2 ,$$

(7)

with

$$\hat{u}_M^\star(x_t,t) = \sum_{j=1}^M \frac{b^{(j)} - x_t}{1-t} \cdot \left[\text{softmax}\left(\left(-\frac{\|x_t - tb^{(k)}\|^2}{2(1-t)^2}\right)_{k=1,\ldots,M}\right)\right]_j .$$

(8)

Crucially, in Equation (7) the sample $b^{(1)}$ depends on $x_t$ and is reused in the estimate $\hat{u}_M^\star$. This important detail yields an unbiased estimator of $\hat{u}^\star$.

### B.2  Theoretical properties of the proposed estimator

First, we discuss below the relation between Proposition 2 and the sampling literature.

**Links with importance sampling**. The estimator $\hat{u}^\star$ in Equation (6) can be seen as a form of *importance sampling* (see Robert et al. 1999, Chap. 3 for an in-depth reference). In a nutshell, importance sampling is a way to estimate an expectation when one cannot easily sample from the random variable it depends on. More precisely, in the ideal case $z \sim p_{\text{data}}$ (as opposed to $z \sim \hat{p}_{\text{data}}$), the velocity field formula is the following

$$u^\star(x_t,t) = \mathbb{E}_{z|x_t,t}\left[u^{\text{cond}}(x_t,z,t)\right]$$

(30)

$$= \int_z u^{\text{cond}}(x_t,z,t)p(z|x_t,t)\mathrm{d}z .$$

(31)

When $z \sim p_{\text{data}}$, it is difficult to sample from $z|x_t, t$, but the latter equation can be rewritten as

$$u^\star(x_t, t) = \int_z u^{\text{cond}}(x_t, z, t) \frac{p(z|x_t, t)}{p(z)} p(z) \mathrm{d}z \tag{32}$$

and one can easily sample from $z \sim \hat{p}_{\text{data}}$ using the empirical data distribution $x^{(1)}, \ldots, x^{(n)}$

$$u^\star(x_t, t) \approx \frac{1}{n} \sum_{i=1}^n u^{\text{cond}}(x_t, x^{(i)}, t) \frac{p(z = x^{(i)}|x_t, t)}{p(x^{(i)})} \tag{33}$$

$$= \sum_{i=1}^n u^{\text{cond}}(x_t, x^{(i)}, t) p(z = x^{(i)}|x_t, t) \tag{34}$$

$$:= \hat{u}^\star(x_t, t) . \tag{35}$$

## B.3    Proof of Proposition 2

We first recall Appendix B, which we prove in this section.

**Proposition 2.** *We denote the conditional probability distribution $p(z = x^{(i)} \mid x, t)$ over $\{x^{(i)}\}_{i=1}^n$ by $\hat{p}_{\text{data}}(z \mid x, t)$. With no constraints on the learned velocity field $u_\theta$,*

*i) The minimizer of Equation (7) writes, for all $(x, t)$*

$$\mathbb{E}_{\substack{b^{(1)} \sim \hat{p}_{\text{data}}(\cdot|x,t) \\ b^{(2)}, \ldots, b^{(M)} \sim \hat{p}_{\text{data}}}} [\hat{u}_M^\star(x, t)] . \tag{9}$$

*ii) In addition, for all $(x, t)$, the minimizer of Equation (7) equals the optimal velocity field, i.e.,*

$$\mathbb{E}_{\substack{b^{(1)} \sim \hat{p}_{\text{data}}(\cdot|x,t) \\ b^{(2)}, \ldots, b^{(M)} \sim \hat{p}_{\text{data}}}} [\hat{u}_M^\star(x, t)] = \hat{u}^\star(x, t) . \tag{10}$$

*iii) The conditional variance of the estimator $\hat{u}_M^\star$ is smaller than the usual conditional variance:*

$$\text{Var}_{\substack{b^{(1)} \sim \hat{p}_{\text{data}}(\cdot|x,t) \\ b^{(2)}, \ldots, b^{(M)} \sim \hat{p}_{\text{data}}}} [\hat{u}_M^\star(x, t)] \leq \text{Var}_{b^{(1)} \sim \hat{p}_{\text{data}}(\cdot|x,t)} \left[ u^{\text{cond}}(x, b^{(1)}, t) \right]. \tag{11}$$

*Proof of Item (i).* With no constraints on $u_\theta$, the empirical flow matching loss writes:

$$\mathbb{E}_{\substack{t \sim \mathcal{U}([0,1]) \\ x_1 \sim \hat{p}_{\text{data}} \\ x_t = (1-t)x_0 + tx_1 \\ b^{(1)} := x_1 \, ; \, b^{(2)}, \ldots, b^{(M)} \sim \hat{p}_{\text{data}}}} \|u_\theta(x_t, t) - \hat{u}_M^\star(x_t, t)\|^2 , \tag{36}$$

$$= \mathbb{E}_{\substack{t \sim \mathcal{U}([0,1]) \\ x_t \sim p_t}} \mathbb{E}_{\substack{b^{(1)} \sim \hat{p}_{\text{data}}(\cdot|x_t,t) \\ b^{(2)}, \ldots, b^{(M)}|x_t, t}} \|u_\theta(x_t, t) - \hat{u}_M^\star(x_t, t)\|^2 , \tag{37}$$

$$= \mathbb{E}_{\substack{t \sim \mathcal{U}([0,1]) \\ x_t \sim p_t}} \mathbb{E}_{\substack{b^{(1)} := \hat{p}_{\text{data}}(\cdot|x_t,t) \\ b^{(2)}, \ldots, b^{(M)} \sim \hat{p}_{\text{data}}}} \|u_\theta(x_t, t) - \hat{u}_M^\star(x_t, t)\|^2 \text{ because } b^{(2)}, \ldots, b^{(M)} \perp\!\!\!\perp x_t, t , \tag{38}$$

which is minimized when for all $x_t, t$

$$u_\theta(x_t, t) = \mathbb{E}_{\substack{b^{(1)} \sim \hat{p}_{\text{data}}(\cdot|x_t,t) \\ b^{(2)}, \ldots, b^{(M)} \sim \hat{p}_{\text{data}}}} [\hat{u}_M^\star(x_t, t)] . \tag{39}$$

$\square$

*Proof of Item (ii).* The minimizer for a given $(x_t, t)$, removing these elements from the notation for conciseness and abstraction, is a weighted mean:

$$\hat{u}^\star(x_t, t) = \hat{u}^\star = \sum_{l=1}^n w^{(l)} u^{(l)} , \text{ with} \tag{40}$$

$$w^{(l)} = \hat{p}_{\text{data}}(z = x^{(l)}|t, x_t) , \sum_{l=1}^n w^{(l)} = 1 \tag{41}$$

$$u^{(l)} = u^{\text{cond}}(x_t, x^{(l)}, t) \tag{42}$$

We express a mini-batch as an $M$-valued vector of indices, $\boldsymbol{i} \in [\![1, n]\!]^M$. The mini-batch estimate from Equation (7), considering the definition of the softmax, can be expressed as a mini-batch weighted-mean:

$$\hat{u}_M^\star(\mathbf{i}) = \frac{\sum_{j=1}^{M} w^{(\mathbf{i_j})} u^{(\mathbf{i_j})}}{\sum_{j=1}^{M} w^{(\mathbf{i_j})}} \tag{43}$$

The categorical distribution over $[\![1, n]\!]$ with probabilities following the weights $w$ in (41) is denoted $\text{Cat}(w)$ and the uniform distribution, *i.e.,* $\text{Cat}(\mathbf{1}/n)$), is denoted $\text{Unif}$.

The main result of the following is that, in expectation over the biased-mini-batches, **where the first point is drawn according to** $w$ and the $M - 1$ other points are drawn uniformly, the mini-batch weighted-mean is an unbiased estimate of the $w$-weighted-mean $\hat{u}^\star$.

$$\mathbb{E}\left[\hat{u}_M^\star(\mathbf{i})\right] := \mathbb{E}_{\boldsymbol{i_1} \sim \text{Cat}(w)} \mathbb{E}_{\boldsymbol{i_2}, \dots, \boldsymbol{i_M} \sim \text{Unif}}\left[\hat{u}_M^\star(\mathbf{i})\right] \tag{44}$$

$$= \sum_{\boldsymbol{i_1}=1}^{n} w^{(\boldsymbol{i_1})} \mathbb{E}_{\boldsymbol{i_2}, \dots, \boldsymbol{i_M} \sim \text{Unif}}\left[\hat{u}_M^\star(\mathbf{i})\right] \tag{45}$$

$$= \sum_{\boldsymbol{i_1}=1}^{n} \mathbb{E}_{\boldsymbol{i_2}, \dots, \boldsymbol{i_M} \sim \text{Unif}}\left[w^{(\boldsymbol{i_1})} \hat{u}_M^\star(\mathbf{i})\right] \tag{46}$$

$$= n \sum_{\boldsymbol{i_1}=1}^{n} \frac{1}{n} \mathbb{E}_{\boldsymbol{i_2}, \dots, \boldsymbol{i_M} \sim \text{Unif}}\left[w^{(\boldsymbol{i_1})} \hat{u}_M^\star(\mathbf{i})\right] \tag{47}$$

$$= n \, \mathbb{E}_{\boldsymbol{i_1} \sim \text{Unif}} \mathbb{E}_{\boldsymbol{i_2}, \dots, \boldsymbol{i_M} \sim \text{Unif}}\left[w^{(\boldsymbol{i_1})} \hat{u}_M^\star(\mathbf{i})\right] \tag{48}$$

$$= n \, \mathbb{E}_{\boldsymbol{i_1}, \dots, \boldsymbol{i_M} \sim \text{Unif}}\left[w^{(\boldsymbol{i_1})} \hat{u}_M^\star(\mathbf{i})\right] \tag{49}$$

The expression in Equation (49) is invariant with respect to order of the indices $\boldsymbol{i}_1, \dots, \boldsymbol{i}_M$: the indices in expectation in Equation (49) can be exchanged, and one thus has

$$\forall k \in [\![1, M]\!], \; \mathbb{E}\left[\hat{u}_M^\star(\mathbf{i})\right] = n \, \mathbb{E}_{\boldsymbol{i_1}, \dots, \boldsymbol{i_M} \sim \text{Unif}}\left[w^{(\boldsymbol{i_k})} \hat{u}_M^\star(\mathbf{i})\right] \; . \tag{50}$$

Averaging Equation (50) over the indices $k \in [\![1, M]\!]$ yields the desired result

$$\frac{1}{M} \sum_{k=1}^{M} \mathbb{E} \hat{u}_M^\star(\mathbf{i}) = \frac{1}{M} \sum_{k=1}^{M} n \, \mathbb{E}_{\boldsymbol{i}_1,\dots,\boldsymbol{i}_M \sim \mathrm{Unif}} \left[ w^{(\boldsymbol{i}_k)} \hat{u}_M^\star(\mathbf{i}) \right] \tag{51}$$

$$\mathbb{E} \hat{u}_M^\star(\mathbf{i}) = \frac{1}{M} n \, \mathbb{E}_{\boldsymbol{i}_1,\dots,\boldsymbol{i}_M \sim \mathrm{Unif}} \left[ \sum_{k=1}^{M} w^{(\boldsymbol{i}_k)} \hat{u}_M^\star(\mathbf{i}) \right] \tag{52}$$

$$= \frac{1}{M} n \, \mathbb{E}_{\boldsymbol{i}_1,\dots,\boldsymbol{i}_M \sim \mathrm{Unif}} \left[ \sum_{k=1}^{M} w^{(\boldsymbol{i}_k)} \frac{\sum_{j=1}^{M} w^{(\mathbf{i_j})} u^{(\mathbf{i_j})}}{\sum_{j=1}^{M} w^{(\mathbf{i_j})}} \right] \tag{53}$$

$$= \frac{1}{M} n \, \mathbb{E}_{\boldsymbol{i}_1,\dots,\boldsymbol{i}_M \sim \mathrm{Unif}} \left[ \left( \sum_{k=1}^{M} w^{(\boldsymbol{i}_k)} \right) \frac{\sum_{j=1}^{M} w^{(\mathbf{i_j})} u^{(\mathbf{i_j})}}{\left( \sum_{j=1}^{M} w^{(\mathbf{i_j})} \right)} \right] \tag{54}$$

$$= \frac{1}{M} n \, \mathbb{E}_{\boldsymbol{i}_1,\dots,\boldsymbol{i}_M \sim \mathrm{Unif}} \left[ \sum_{j=1}^{M} w^{(\mathbf{i_j})} u^{(\mathbf{i_j})} \right] \tag{55}$$

$$= \frac{1}{M} n \sum_{j=1}^{M} \mathbb{E}_{\boldsymbol{i}_1,\dots,\boldsymbol{i}_M \sim \mathrm{Unif}} \left[ w^{(\mathbf{i_j})} u^{(\mathbf{i_j})} \right] \tag{56}$$

$$= \frac{1}{M} n \sum_{j=1}^{M} \mathbb{E}_{\boldsymbol{i}_j \sim \mathrm{Unif}} \left[ w^{(\mathbf{i_j})} u^{(\mathbf{i_j})} \right] \tag{57}$$

$$= \frac{1}{M} n \, M \, \mathbb{E}_{l \sim \mathrm{Unif}} \left[ w^{(l)} u^{(l)} \right] \tag{58}$$

$$= n \, \mathbb{E}_{l \sim \mathrm{Unif}} \left[ w^{(l)} u^{(l)} \right] \tag{59}$$

$$= n \sum_{l=1}^{n} \frac{1}{n} \left[ w^{(l)} u^{(l)} \right] \tag{60}$$

$$= \sum_{l=1}^{n} \left[ w^{(l)} u^{(l)} \right] \tag{61}$$

$$= \hat{u}^\star \tag{62}$$

$\square$

*Proof of Item* (iii)). Using the same ideas as for Item (ii)), one has

$$\mathbb{E}_{x^{(1)} \sim \hat{p}_{\text{data}}(\cdot | x_t, t) \, ; \, b^{(2)}, \ldots, b^{(M)} \sim \hat{p}_{\text{data}}} \left[ \hat{u}_M^\star(x_t, t)^2 \right] \tag{63}$$

$$= n \mathbb{E}_{\boldsymbol{i}_1, \ldots, \boldsymbol{i}_M \sim \text{Unif}} \left[ w^{(\boldsymbol{i}_1)} \hat{u}_M^\star(\boldsymbol{i})^2 \right] \tag{64}$$

$$= n \mathbb{E}_{\boldsymbol{i}_1, \ldots, \boldsymbol{i}_M \sim \text{Unif}} \left[ w^{(\boldsymbol{i}_k)} \hat{u}_M^\star(\boldsymbol{i})^2 \right], \forall k \in [\![ 1, M ]\!] \tag{65}$$

$$= n \frac{1}{M} \mathbb{E}_{\boldsymbol{i}_1, \ldots, \boldsymbol{i}_M \sim \text{Unif}} \left[ \sum_{k=1}^{M} w^{(\boldsymbol{i}_k)} \hat{u}_M^\star(\boldsymbol{i})^2 \right] \tag{66}$$

$$= n \frac{1}{M} \mathbb{E}_{\boldsymbol{i}_1, \ldots, \boldsymbol{i}_M \sim \text{Unif}} \left[ \sum_{k=1}^{M} w^{(\boldsymbol{i}_k)} \left( \frac{\sum_{j=1}^{M} w^{(\boldsymbol{i}_j)} u^{(\boldsymbol{i}_j)}}{\sum_{j=1}^{M} w^{(\boldsymbol{i}_j)}} \right)^2 \right] \tag{67}$$

$$\leq n \frac{1}{M} \mathbb{E}_{\boldsymbol{i}_1, \ldots, \boldsymbol{i}_M \sim \text{Unif}} \left[ \sum_{k=1}^{M} w^{(\boldsymbol{i}_k)} \frac{\sum_{j=1}^{M} w^{(\boldsymbol{i}_j)} (u^{(\boldsymbol{i}_j)})^2}{\sum_{j=1}^{M} w^{(\boldsymbol{i}_j)}} \right] \text{ by convexity of } x \mapsto x^2 \tag{68}$$

$$= n \frac{1}{M} \mathbb{E}_{\boldsymbol{i}_1, \ldots, \boldsymbol{i}_M \sim \text{Unif}} \left[ \left( \sum_{k=1}^{M} w^{(\boldsymbol{i}_k)} \right) \frac{\sum_{j=1}^{M} w^{(\boldsymbol{i}_j)} (u^{(\boldsymbol{i}_j)})^2}{\sum_{j=1}^{M} w^{(\boldsymbol{i}_j)}} \right] \tag{69}$$

$$= n \frac{1}{M} \mathbb{E}_{\boldsymbol{i}_1, \ldots, \boldsymbol{i}_M \sim \text{Unif}} \left[ \sum_{j=1}^{M} w^{(\boldsymbol{i}_j)} (u^{(\boldsymbol{i}_j)})^2 \right] \tag{70}$$

$$= \mathbb{E}_{\boldsymbol{i}_1 \sim \text{Unif}} \left[ w^{(\boldsymbol{i}_1)} (u^{(\boldsymbol{i}_1)})^2 \right] \tag{71}$$

$$= \mathbb{E}_{l \sim \text{Unif}} \left[ w^{(l)} (u^{(l)})^2 \right] . \tag{72}$$

Hence

$$\mathbb{E}_{x^{(1)} \sim \hat{p}_{\text{data}}(\cdot | x_t, t) \, ; \, b^{(2)}, \ldots, b^{(M)} \sim \hat{p}_{\text{data}}} \left[ \hat{u}_M^\star(x_t, t)^2 \right] - (\hat{u}^\star)^2 \leq \mathbb{E}_{l \sim \text{Unif}} \left[ w^{(l)} (u^{(l)})^2 \right] - (\hat{u}^\star)^2 \; , \tag{73}$$

which is exactly

$$\text{Var}_{x^{(1)} \sim \hat{p}_{\text{data}}(\cdot | x_t, t) \, ; \, b^{(2)}, \ldots, b^{(M)} \sim \hat{p}_{\text{data}}} \left[ \hat{u}_M^\star(x_t, t) \right] \leq \text{Var}_{x^{(1)} \sim \hat{p}_{\text{data}}(\cdot | x_t, t)} \left[ u^{\text{cond}}(x_t, x^{(1)}, t) \right] \; . \tag{74}$$

$\square$

## C  Additional experiments

We present below the results for the MNIST dataset. The conclusions atre the same as for the CIFAR-10 and CelebA $64 \times 64$: regressing against a more deterministic velocity field does not hurt generalization. On the contrary, generalization (*i.e.,* lower test FID) appears earlier during training.

For this experiment, we used the Unet with attention and timestep embedding from `torchcfm` library, with the Adam optimizer and all the default parameters. We used a pretrained classifier with 99% accuracy on MNIST (90% on FMNIST) as a lower-dimensional embedding of size 128 to compute the FID between the test set and the generated set.

| Method | Ep. 1 | Ep. 2 | Ep. 3 | Ep. 4 | Ep. 5 | Ep. 10 | Ep. 15 | Ep. 20 | Ep. 25 |
|---|---|---|---|---|---|---|---|---|---|
| CFM (EFM, M=1) | 378.00 | 181.25 | 67.88 | 29.44 | 15.30 | 4.20 | 3.08 | 2.51 | 2.28 |
| EFM, M=128 | 370.64 | 168.58 | 60.52 | 25.52 | 13.44 | 3.79 | 2.70 | 2.35 | 2.10 |
| EFM, M=256 | 370.94 | 169.71 | 61.88 | 25.73 | 13.48 | 3.73 | 2.76 | 2.33 | 2.08 |
| EFM, M=1024 | 369.72 | 168.43 | 60.28 | 24.24 | 12.26 | 3.30 | 2.67 | 2.17 | 1.84 |

Table 1: **FID FMNIST**. FID scores across training epochs for conditional flow matching and empirical flow matching for multiple values of the number of samples $M$ used to estimate the closed-form $\hat{u}^\star$.

| Method | FID Ep. 5 | FID Ep. 10 | FID Ep. 50 | FID Ep. 100 | FID Ep. 200 |
|---|---|---|---|---|---|
| CFM (EFM, M=1) | 253.56 | 48.67 | 25.36 | 21.35 | 19.67 |
| EFM, M=128 | 206.27 | 44.08 | 23.39 | 19.63 | 17.72 |
| EFM, M=256 | 202.62 | 45.06 | 22.16 | 20.08 | 17.74 |
| EFM, M=512 | 194.66 | 44.19 | 22.10 | 18.93 | 16.85 |

Table 2: **FID FMNIST**. FID scores across training epochs for conditional flow matching and empirical flow matching for multiple values of the number of samples $M$ used to estimate the closed-form $\hat{u}^\star$.

# D  Experiments details

For all the experiment we used all the same learning hyperparameters, the default ones form Tong et al. (2024). The hyperparameter values are summarized in Table 3. The details specific to each figure are described in Appendices D.2 to D.5

| # Channels | Batch Size | Learning Rate | EMA Decay | Gradient Clipping |
|---|---|---|---|---|
| 128 | 128 | 0.0002 | 0.9999 | 1 |

Table 3: Learning hyperparameters for all the CIFAR-10 and CelebA $64$ experiments.

## D.1  Compute time

Given that regressing against an estimate of the closed-form, EFM, seems to improve on CFM, one may wonder what is the additional cost induced by EFN. To alleviate the non-linearity of GPU computing (parallelism may cause some discontinuities in terms of costs), we ran an exhaustive set of timing experiments, varying the batch size and the EFM sample size. To summarize the measurements (numbers are given for an NVIDIA L4 GPU, on CIFAR-10), denoting $b$ the batch size and $e$ the EFM sample size, the cost follows $b \times (4.3ms + e \times 0.9\mu s)$. It can be also be seen as adding $\sim 2\%$ for every 100 EFM samples. Or, for instance with a batch size of 256, $1.1$ second will be due to the 256-sample forward/backward, while the additional cost for EFM-1000 will be 230ms (around 17% of the cost) and for EFM-128 under 30ms (under 3%).

## D.2  Figures 1a and 1c

For Figure 1a no deep learning is involved: the datasets 2-moons and CIFAR-10 are loaded. Then, 256 points from $p_0 \times \hat{p}_{\text{data}}$ are drawn, and one computes the mean of the cosine similarities between $\hat{u}^\star((1-t)x_0 + tx_1, t)$ and $u^{\text{cond}}((1-t)x_0 + tx_1, z = x_1, t) = x_1 - x_0$, for each value of $t \in \{0, 1/100, 2/100, \ldots, 99/100\}$.

No deep learning either is involved in Figure 1c: the Imagenette dataset is loaded and spatially subsampled to resolution $\dim = 8$, $\dim = 16$, ..., $\dim = 256$, *i.e.,* with $d = " \cdot 8^2$, $d = 3 \cdot 16^2$, ..., $d = 3 \cdot 256^2$. Then, as for Figure 1a, batches of 256 points from $p_0$ and $p_{\text{data}}$ are drawn, and one computes the percentage of cosine similarities between $\hat{u}^\star((1-t)x_0 + tx_1, t)$ and $u^{\text{cond}}((1-t)x_0 + tx_1, z = x_1, t) = x_1 - x_0$, that are larger than $0.9$, for multiple time values $t$.

## D.3  Figure 2

In Figure 2, networks are trained with a vanilla conditional flow matching, with the standard 34 million parameters U-Net for diffusion by Nichol and Dhariwal (2021), with default settings from the `torchfm` codebase [4] (Tong et al., 2024). Training uses the CFM loss. For this specific experiment, **we removed the usual random flip transform**, for $\hat{u}^\star$ to be simpler and easier to estimate by $u_\theta$. For each "data" subsampling of the dataset, we trained the model for $5 \cdot 10^4$ iterations, with a batch size of 128, *i.e.,* we trained the models for 128 epochs.

---

[4] https://github.com/atong01/conditional-flow-matching

### D.4 Figure 3

In Figure 3, for each dataset (CIFAR-10 and CelebA $64 \times 64$), one network is trained using a vanilla conditional flow matching with the default parameters of Tong et al. (2024) (the most important ones are recalled in Table 3). Then images are generated first following the closed-form formula of the optimal velocity field $\hat{u}^\star$ from $0$ to $\tau$. And then following the velocity field learned with a usual conditional flow matching $u_\theta$ from $\tau$ to $1$.

### D.5 Figure 4

For experiments involving training on CIFAR-10 (Figures 2 and 3), we rely on the standard 34 million parameters U-Net for diffusion by Nichol and Dhariwal (2021), with default settings from the `torchfm` codebase (Tong et al., 2024). For each algorithm, the networks are trained for 500k iterations with batch size 128, *i.e.,* 1280 epochs.

For CelebA $64 \times 64$ (Figure 3), we rely on the training script of `pnpflow` library[5] (Martin et al., 2025), which uses a U-Net from Huang et al. (2021); Ho et al. (2020).

---

[5]https://github.com/annegnx/PnP-Flow

