# OpenReview forum: "On the Closed-Form of Flow Matching: Generalization Does Not Arise from Target Stochasticity"
_NeurIPS.cc/2025/Conference — NeurIPS 2025 oral_

### Official Review · Reviewer_xahj · 2025-06-30

**Clarity:** 2
**Significance:** 4
**Originality:** 4
**Rating:** 5
**Confidence:** 5

**Summary:**

This manuscript investigates whether the stochasticity introduced by the conditional flow matching loss contributes to model generalization. While prior work suggests that conditioning on vector fields induces beneficial randomness, the authors demonstrate that, in high-dimensional settings, this stochastic effect occurs only within a narrow time window, casting doubt on its generalization benefit. To address this, they propose a novel training algorithm that interpolates between conditional and unconditional flow matching losses, empirically validating that even limited stochasticity can enhance generalization performance.

**Questions:**

1. In line 89, the notation $z\sim p_0\times\hat{p_{data}}$ is ambiguous. Do you intend that z is sampled from $\hat{p}_{data}$ while an independent noise term is drawn from $p_0$?
2. Figure 5 is difficult to interpret: the gray curve is nearly invisible, and the reported gains from EFM appear marginal. I recommend reformatting the results—perhaps via a table or clearer color scheme—to improve readability.
3. Regarding the loss in Equation (7), can it be viewed as an interpolation between the conditional and unconditional flow matching losses? In particular, setting M=1 recovers the conditional loss, while M=the number of data seems to correspond to the unconditional loss.

**Ethical Concerns:**

["NO or VERY MINOR ethics concerns only"]

**Final Justification:**

I had a misinterpretation on the purpose of section 4 during my review. The authors pointed this out and I agree with what they said. They have hence resolved my biggest concern. I like the results of this paper as it provides a concise answer to an issue raised not long ago and this paper may be helpful for the community in understanding the generalization of flow matching.

**Limitations:**

yes

**Quality:**

3

**Strengths And Weaknesses:**

**Strengths**

- The paper offers compelling empirical insights into how conditional vector fields differ from their unconditional counterparts, deepening our understanding of flow matching training dynamics.
- The proposed interpolation framework between loss functions is novel and may inspire future theoretical developments.

**Weaknesses**

- The manuscript’s narrative lacks coherence: the title and early claims argue against stochasticity as a driver of generalization, yet Section 4 appears to contradict this by showing that reducing stochasticity can improve performance. A unified, logically consistent storyline is needed.

---

> ### Author Rebuttal · Authors · 2025-07-31
>
> We would to thank Reviewer xahj for the very positive feeback, highlighting the compelling empirical analysis. Below, we provide answer to the specific questions.
>
> > The title and early claims argue against stochasticity as a driver of generalization, yet Section 4 appears to contradict this by showing that reducing stochasticity can improve performance.
>
>
> We are confused by this sentence; in our opinion **both the title and Section 4 consistently support the same message**:
> - **The title argues against target stochasticity** as a key driver of generalization
> - **Section 4 supports that target stochasticity does not bring generalization improvement** compared to its more deterministic counterpart. Our findings tend to support that regressing against a deterministic target (EFM, Fig 5) yields generalization earlier during training.  In particular, Section 4 shows that the more deterministic the target (i.e., the larger the number of samples $M$ to estimate the closed-form formula), the earlier the generalization occurs.
>
> We would be grateful if the reviewer could elaborate on which part of the narrative appeared contradictory, so we can address it more explicitly in the revised manuscript.
>
>
> > 1. In line 89, the notation “$z \sim p_0 \times \hat p_\mathrm{data}$” is ambiguous. Do you intend that $z$ is sampled from $\hat p_\mathrm{data}$ while an independent noise term is drawn from $p_0$ ?
>
>
>  What we meant here is that instead of choosing as conditioning variable $z = x^{(i)} \sim \hat p_\mathrm{data}$, an alternate choice (used eg by Albergo and Vanden Eijnden 2023, or Liu et al 2023) is to use $z = (x_0, x^{(i)})$ where $x_0$ is noise. This sentence has been removed for clarity.
>
>
> > 2. Figure 5 is difficult to interpret: the gray curve is nearly invisible, and the reported gains from EFM appear marginal. I recommend reformatting the results—perhaps via a table or clearer color scheme—to improve readability.
>
> Readability has been improved using larger figures. However, we would like to underscore that the goal of this figure is not to show that "Empirical flow matching (EFM) works better" than conditional flow matching. The core idea is to show that the estimator based on the stochastic target (vanilla conditional flow matching) and the estimator with a more deterministic target (empirical flow matching) yield very similar generalization performance. Hence, **target stochasticity is not the driver of generalization.**
>
> In addition, as the reviewer observed the curves for $M=256$ and $M=1000$ are almost identical, which shows that above this threshold, using more points in EFM does not affect the performance. We have made this clearer in the discussion of the experiments.
>
> > 3. Regarding the loss in Equation (7), can it be viewed as an interpolation between the conditional and unconditional flow matching losses? In particular, setting M=1 recovers the conditional loss, while M=the number of data seems to correspond to the unconditional loss.
>
> Yes! This is exactly the idea!
> - $M=1$ recovers the vanilla conditional flow matching loss, with a stochastic target.
> - $M = \text{number of samples}$ recovers fully the flow matching loss, with a deterministic target, that would be very costly to compute in practice
> - $1 < M < \text{number of samples}$ is a stochastic, but more deterministic target that is cheaper to compute in practice ($\approx 1$% overhead for $M=1000$). Please also refer to the discussion with Reviewer Byfq for additional details on the timings.

---

> > ### Comment · Reviewer_xahj · 2025-08-03
> >
> > Thank you for addressing my concerns.
> >
> > For the notation in line 89, I do believe that writing explicitly $z = (x_0, x^{(i)})$ will make things clearer.
> >
> > I realized that I've misinterpreted section 4 and I agree with the authors that it validates the point that stochasticity is not contributing to generalization. I will adjust my score accordingly.

---

> > > ### Author Response · Authors · 2025-08-04
> > >
> > > Thank you again for your thoughtful engagement and for considering our clarifications. Just checking in kindly, if you're still planning to update your score. Please let us know if any points remain unclear.

---

### Official Review · Reviewer_ik8M · 2025-07-03

**Clarity:** 3
**Significance:** 3
**Originality:** 3
**Rating:** 5
**Confidence:** 3

**Summary:**

The paper analyses the reasons for generalization of flow matching generative models. Its main focus is ruling out a recent explanation identifying the stochasticity of the target as the root for generalization of flow matching. The authors make use of the closed form optimal solution of the velocity field to eliminate the stochasticity and show that although the target is deterministic, the model still generalizes (and even surprisingly performs better).

**Questions:**

- Can the authors extend the discussion on how the error in approximating the velocity relates to the dynamical regimes found in [\[2402.18491\] Dynamical Regimes of Diffusion Models](https://arxiv.org/abs/2402.18491) ?

**Ethical Concerns:**

["NO or VERY MINOR ethics concerns only"]

**Final Justification:**

I keep my originally given high score for this paper and recommend acceptance.

**Limitations:**

yes

**Quality:**

3

**Strengths And Weaknesses:**

**Strengths**

- Clarity: the paper is well written, the problem and the experimental settings are clear.
- Originality: the paper takes a courageous approach challenging another work’s conclusions and it does so respectfully and backed up with thorough experimental results.
- Removing loss stochasticity could simplify implementation and improve Flow Matching model training stability/efficiency.

**Weaknesses**

- The analysis is done only over image modality and with U-Net architecture. Would be interesting to see if the same behavior happens for other modalities/architectures and how this affects the error in approximating the velocity.

---

> ### Author Rebuttal · Authors · 2025-07-31
>
> We would like to thank Reviewer ik8M for the very positive feedback and for highlighting the clarity and originality of the manuscript. Below, we answer the specific questions.
>
> > The analysis is done only over image modality and with U-Net architecture. Would be interesting to see if the same behavior happens for other modalities/architectures and how this affects the error in approximating the velocity.
>
> We agree with the reviewer that our paper specifically targets image generation, as this is the main application for Flow Matching. In all the experiments, we used the reference Unet with attention and timestep embedding from the reference torchcfm library.
> We believe that we would obtain the same results with other architectures : for diffusion, [1] have shown that multiple architectures (DiT, U-ViT DDPM++, NCSN) end up learning the same velocity field (see their Fig. 1 and Fig 9)
>
> [1] Towards a Mechanistic Explanation of Diffusion Model Generalization, ICML 2025, Niedoba, Matthew and Zwartsenberg, Berend and Murphy, Kevin and Wood, Frank
>
>
> > Can the authors extend the discussion on how the error in approximating the velocity relates to the dynamical regimes found in "Dynamical Regimes of Diffusion Models" ?
>
> - The analysis in "Dynamical Regimes of Diffusion Models" [Biroli et al., 2024] focuses on the behavior of the **closed-form score function** in diffusion models, **without considering how neural networks actually learn or approximate this function**. Their main contribution is to identify three dynamical regimes—noisy, speciation, and collapse—that emerge when sampling is performed using the exact, analytical score in a Gaussian mixture setting. These regimes are derived from connections to classical phase transitions in statistical physics.
> - In contrast, our work investigates **how deep learning models approximate this ideal score function**, and more importantly, **how this failure to approximate the "optimal" score impact generalization**. We empirically observe that model errors are concentrated around two critical timesteps (Figure 3), which may correspond to key dynamical transitions.
> - Regarding the study of the non-stochasticity of the velocity field, we added experiments showing how the critical time for non-stochasticity (defined as a cosine similarity between $u^\text{cond}$ and $\hat u$ greater than 0.9) varies with both the number of training samples $n$ and  the data dimension $d$. We observe a dependence in $\frac{\log n}{d}$, which is the same as the one obtained by Biroli for their 'collapse time'  (defined as the time from which the diffusion trajectories are attracted by a single training point).
>
> Drawing connections between the empirical failure modes we observe and the theoretical regimes identified by Biroli et al. is a promising direction for future work. Such a connection would require extending their theoretical framework to account for learning dynamics and approximation errors introduced by neural architectures.

---

> > ### Comment · Reviewer_ik8M · 2025-08-05
> >
> > I thank the authors for their response and effort is addressing my questions.
> >
> > > Regarding the study of the non-stochasticity of the velocity field, we added experiments showing how the critical time for non-stochasticity (defined as a cosine similarity between $u^\text{cond}$ and $\hat u$ greater than 0.9) varies with both the number of training samples $n$ and the data dimension $d$. We observe a dependence in $\frac{\log n}{d}$, which is the same as the one obtained by Biroli for their 'collapse time' (defined as the time from which the diffusion trajectories are attracted by a single training point).
> >
> > I find this finding especially interesting! how might the time in which there are larger approximation errors in the learned flow model may connect to these generation regimes and their connections to generalization.
> >
> > I recommand acceptance of tha peper.

---

### Official Review · Reviewer_Byfq · 2025-07-03

**Clarity:** 3
**Significance:** 2
**Originality:** 3
**Rating:** 5
**Confidence:** 3

**Summary:**

This paper targets explaining the source of generalization in flow matching related methods.
The authors argue that it is the noisy loss, rather than the noisy data, that contribute to the generalization ability in flow matching.
The authors conducted interesting experiments to investigate the behavior of the target velocity as a function of time, and conclude that the introducing randomness in the training loss would lead to better generalization.
Inspired by this idea, the authors proposed a novel flow matching loss called empirical flow matching (EFM) and performed experiments to verify the generalization ability of EFM.

**Questions:**

Please see the "strength and weakness" part.

**Ethical Concerns:**

["NO or VERY MINOR ethics concerns only"]

**Final Justification:**

The authors fully addressed my concerns, so I'd like to raise my score to 5.

**Limitations:**

The authors should discuss their computational cost with empirical evidences.

**Quality:**

2

**Strengths And Weaknesses:**

## Strengths
- Towards understanding the generalization ability of flow matching, this paper provides a very interesting perspective on the learned velocity as a weighted sum of the conditional velocities and analyze the behavior of the weights.
- I find the empirical validations in Sec 3 make sense for explaining the behavior of the velocity and the resulting generalization ability,
- The novel EFM loss function is also deemed as a major contribution of this paper.

## Weaknesses
- This paper does not give a clear mathematical definition for the "generalization" and build their frameworks purely on (somewhat vague) empirical results. For example, can the generalization ability be defined as the velocity matching error on the training set? So that the relationship between the generalization ability and the training set size / piecewise threshold can be discussed more mathematically.
-  I agree with the authors that the $M$ additional samples can be efficiently tackled by vectorization and the same backpropagation depth. However, empirical evidence on computational time and memory should be provided, as EFM explicitly introduces additional costs.
- The experiments are not sufficient to prove the superiority of EFM, as different methods almost converge as the training proceeds. Moreover, to the best of my knowledge, quantititive results for flow matching methods can have large randomness.
- In my opinion, the generalization ability generally leads to more evidently better test results than training results. However, in Figure 5, the training results and test results are almost the same. So can the authors explain this phenomenon?

---

> ### Author Rebuttal · Authors · 2025-07-31
>
> We would like to thank Reviewer Byfq for the very positive feedback and highlighting the new perspectives on generalization provided by the paper. Below, we answer the specific concerns.
>
> > This paper does not give a clear mathematical definition for the "generalization" and build their frameworks purely on (somewhat vague) empirical results. For example, can the generalization ability be defined as the velocity matching error on the training set?
>
> Mathematically defining and measuring the generalization of generative models is a challenging and still largely open problem. As per [1], there are three criteria: “quantifying to what extent images resemble those from the training set (fidelity), how well the generated samples span the full training distribution (diversity), and whether they are truly novel or are simply reproductions of training samples (memorization)”.
>
> Standard evaluation metrics like FID have documented flaws [1,2]. For ourselves, we aimed to go beyond standard practice by:
> - Using **more expressive and discriminative embeddings** (DINOv2) instead of the commonly used Inception features
> - Measuring (approximate) Wasserstein distance between the representations of the **generated set and the test set**, rather than the training set, as it is usual done in generative modeling benchmarks
> - Displaying a **memorization metric** that would detect a pure copy of the training set
>
> We have extended the discussion in the Metrics paragraph L244 to clarify this discussion.
>
> > I agree with the authors that the additional samples can be efficiently tackled by vectorization and the same backpropagation depth. However, empirical evidence on computational time and memory should be provided, as EFM explicitly introduces additional costs.
>
> - We emphasize that our main contribution is to challenge target stochasticity as the main driver of generalization: the main point of Section 4 is that the more deterministic EFM works at least as well as the stochastic CFM.
> -  However, given that EFM turns out to improve on CFM, its cost indeed becomes very interesting. To alleviate the non-linearity of GPU computing (parallelism may cause some discontinuities in terms of costs), we ran an exhaustive set of timing experiments, varying the batch size and the EFM sample size. To summarize the measurements (numbers are given for an NVIDIA L4 GPU, on CIFAR-10), denoting $b$ the batch size and $e$ the EFM sample size, the cost follows approximately $b \times ( 4.1\mathrm{ms} + e \times 0.04 \mathrm{μs})$. It can be also be seen as adding ~1% for every 1000 EFM samples. Or, for instance with a batch size of 256, 1.05 sec will be due to the 256-sample forward/backward, while the additional cost for EFM-1000 will be 10ms (around 1%) and for EFM-128, under 1.3 ms.
>
> > The experiments are not sufficient to prove the superiority of EFM, as different methods almost converge as the training proceeds. Moreover, to the best of my knowledge, quantititive results for flow matching methods can have large randomness.
>
> We agree with the reviewer, and we emphasize that **our claim is not that the EFM should become the new standard paradigm for training**. We only mean to show that, when target stochasticity is removed from the problem (as in the EFM objective), models still generalize well, reaching as low FID as other training methods. This rules out target stochasticity as the driver for generalization. We added MNIST and will add FMNIST and tiny_imagenet by the end of the reviewers-authors discussion period, to strengthen this statement (see answer to reviewer MQbh).
>
> > In my opinion, the generalization ability generally leads to more evidently better test results than training results. However, in Figure 5, the training results and test results are almost the same. So can the authors explain this phenomenon?
>
> In all our experiments, **we indeed observe a strong correlation between FID scores on the training and test sets**. This suggests that the models generalize well and that there is no significant overfitting in terms of FID. We believe this strong correlation between train and test FID may also reflect a limitation of the FID metric itself: FID does not penalize overfitting and does not detect subtle generalization gaps. This has been discussed in prior work (e.g., [2]).
>
> While the close alignment between train and test FID may initially seem surprising, it is consistent with prior observations in generative model evaluation and reinforces the practice of using train FID as a proxy for test performance, though, as always, with some caution.
>
>
> **[1]** Stein, G., Cresswell, J., Hosseinzadeh, R., Sui, Y., Ross, B., Villecroze, V., Liu, Z., Caterini, A.L., Taylor, E. and Loaiza-Ganem, G., 2023. Exposing flaws of generative model evaluation metrics and their unfair treatment of diffusion models. Neurips.
>
> **[2]** Jiralerspong, M., Bose, J., Gemp, I., Qin, C., Bachrach, Y. and Gidel, G., 2023. Feature likelihood divergence: evaluating the generalization of generative models using samples. Neurips

---

> > ### Author Response · Authors · 2025-08-04
> >
> > We would greatly appreciate your perspective on the points we addressed in the rebuttal. Please let us know if there are any remaining concerns we can help clarify.

---

> > > ### Author Response · Authors · 2025-08-07
> > >
> > > As the author-reviewer discussion period is coming to a close, we would be happy to clarify any remaining concerns.
> > >
> > > Thank you again for your time and feedback.

---

> > ### Comment · Reviewer_Byfq · 2025-08-08
> >
> > Thanks you for your detailed response. I still suggest adding the empirical time cost discussion to your manuscript. I will keep my positive rating on this paper.

---

> > > ### Author Response · Authors · 2025-08-08
> > >
> > > Thank you for your feedback. We will indeed include the time cost discussion in the paper, as it provides useful information about EFM. If you are satisfied with our answers to your comments and there are no more points to clarify, would you please consider upgrading like your score?

---

### Official Review · Reviewer_MQbh · 2025-07-07

**Clarity:** 4
**Significance:** 2
**Originality:** 2
**Rating:** 5
**Confidence:** 3

**Summary:**

Flow-matching models train a neural network to predict a time-dependent velocity field that carries data points to noise and, in practice, for some unknown reason, they are able to easily generate novel samples. This paper asks why that happens and draws upon two common explanation: first, that the injected noise in the training objective (target stochasticity) forces the model to see many noisy variants and thereby generalize; second, that the neural network cannot exactly fit the closed-form (or memorization) solution for the optimal velocity field, so it learns a smoother field that naturally interpolates between data points.
The authors use the fact that, under perfect fitting, the optimal velocity field is available in closed form and the learned distribution will match the empirical data distribution. To do so, the authors train flow-matching networks and measure how closely the learned field matches the closed-form one at different timeframes. The authors also provide a new training loss that effectively tries to remove the target stochasticity in the flow matching loss and see that no difference occurs in the generalization capabilities of the model. Through these controlled experiments, the authors claim that the generalization does not occur because of target stochasticity.

On the other hand, the authors have an interesting experiment showing that with different dataset sizes, as one increase the size of the dataset a flow-matching model is trained on, while the difference between the closed form velocity field and the learned velocity field increases (the model becomes less and less of a perfect fit) the model obtains better sample quality and novelty. This in turn hints at the fact that the neural network, while not learning the optimal velocity field, learns something (possibly smoother) that generalizes and interpolates better.
Although the paper does not fully explain why the neural parameterization prefers those smoother solutions, its experiments convincingly dismiss the target stochasticity-based hypothesis and underscore the importance of inductive bias in flow-matching models. The work is well written, the experiments are carefully designed, and the insights are strong enough, in my view, to justify acceptance.

**Questions:**

1. In L38, what is meant by the “deterministic regime” of the flow matching objective? Do the authors mean the time steps closer to the data distribution where the generated sample does not change that much?
2. Can you please add the ticks for the x-axis in Figure 3 (the bar plots where each bar indicates the number of samples)

**Ethical Concerns:**

["NO or VERY MINOR ethics concerns only"]

**Final Justification:**

All of the explanations provided seem reasonable and adequately address my questions. After considering these along with the discussions in the other threads, I have raised my score from borderline accept to accept.

**Limitations:**

Highlighted in the weakness section.

**Paper Formatting Concerns:**

The formatting is compliant!

**Quality:**

4

**Strengths And Weaknesses:**

## Strengths

1. Figures 1, 3, and 4 clearly illustrate the controlled experiments, with well-explained setups and convincing results that support the paper’s claims.
2. The authors address the recently identified limitations of FID by using DinoV2 embeddings for evaluation, demonstrating attention to detail.
3. The hybrid model that combines the closed-form solution with a learned neural network is a novel and interesting approach for probing the behaviour of flow-matching models (see Figure 4). The figure was also quite insightful as it showed that the generalization indeed happens early in the generation process, as switching the model to the closed-form version of the velocity field at a threshold time of as little as $\tau=0.3$ removes the generalization capability (this is also consistent with Figure 3).

## Weaknesses
1. While the paper highlights that generalization may arise from the “mismatch” between the neural network and the closed-form velocity field, it does not provide insight into the underlying inductive biases that make this mismatch beneficial. It is already known that the closed-form solution yields a perfectly fitted flow-matching model that memorizes and regurgitates the training data, making it ineffective for generalization. What would be more insightful is an exploration of which types of mismatch are beneficial for generalization, rather than simply observing that a mismatch exists.
2. Section 4 changes the training loss to effectively remove the stochasticity in the training objective. While I do agree that the empirical flow matching loss is “less stochastic,” it is not at all deterministic in any sense. It still relies on a Monte Carlo approximation and I do not believe this experiment alone fully obviates target stochasticity as an explanation for generalization (although I think the rest of the results complement this and the explanations are intuitive and convincing overall). Can you please add an experiment showing training scenarios where the batch hyper parameter in the EFM loss is also swept over? That would effectively control how “stochastic” the Monte-Carlo approximation, and as a result, the loss is.
3. I would be interested in seeing similar experiments not just for CIFAR-10 and CelebA, but also for MNIST, FMNIST, and TinyImagenet to ensure that the conclusions are general enough.

---

> ### Author Rebuttal · Authors · 2025-07-31
>
> We would like to thank Reviewer MQbh for the very positive feedback and for highlighting the clarity and care given the experiments. Below, we answer the specific questions.
>
> > 1. While the paper highlights that generalization may arise from the “mismatch” between the neural network and the closed-form velocity field, it does not provide insight into the underlying inductive biases that make this mismatch beneficial. It is already known that the closed-form solution yields a perfectly fitted flow-matching model that memorizes and regurgitates the training data, making it ineffective for generalization. What would be more insightful is an exploration of which types of mismatch are beneficial for generalization, rather than simply observing that a mismatch exists.
>
> There are two important ideas here. As the reviewer mentions, a closed-form solution exists that only generates training data. Thus, models that generalize fail to “match” this solution.
> - Our **first contribution is to discard “target stochasticity” suggested by Vastola (2025) as a cause for this mismatch**: we show that this phenomenon is limited in realistic high-dimensional settings (Section 3.1), and that models still generalize well when target stochasticity is removed from the problem (Section 4).
> - As a **second contribution, we investigate "which part" of the mismatch is a key driver of generalization**. Section 3.2 highlights "mismatches" at two specific times $t$: $t \approx 0.1$ and $t \approx 1$. Section 3.3 shows that the mismatch at time $t \approx 0.1$ is especially important for generalization: we conclude that failure to learn the optimal velocity at early time steps is the main driver of generalization.
>
>
> > 2. Section 4 changes the training loss to effectively remove the stochasticity in the training objective. While I do agree that the empirical flow matching loss is “less stochastic,” it is not at all deterministic in any sense. It still relies on a Monte Carlo approximation and I do not believe this experiment alone fully obviates target stochasticity as an explanation for generalization (although I think the rest of the results complement this and the explanations are intuitive and convincing overall). Can you please add an experiment showing training scenarios where the batch hyper parameter in the EFM loss is also swept over?
>
> We did not display the experiment with larger batches for the target, because our **results showed that larger batches $M > 1000$ and beyond yield almost the same generalization performances**, and the curves  $M > 1000$ almost superpose. We have added the corresponding parameter sweep in the revised manuscript.
>
> > 3. I would be interested in seeing similar experiments not just for CIFAR-10 and CelebA, but also for MNIST, FMNIST, and TinyImagenet to ensure that the conclusions are general enough.
>
> We present below the results for the MNIST dataset. The conclusion is the same as for the other datasets: **regressing against a more deterministic velocity field does not hurt generalization**. On the contrary, generalization (i.e., lower test FID) appears earlier during training.
>
> For this experiment, we used the Unet with attention and timestep embedding from torchcfm library, with the Adam optimizer and all the default parameters. We used a pretrained classifier with 99% accuracy on MNIST as a lower-dimensional embedding of size 128 to compute the FID between the test set and the generated set.
>
> | Method                           | FID Epoch 1 | FID Epoch 2 | FID Epoch 3 | FID Epoch 4 | FID Epoch 5 | FID Epoch 10 | FID Epoch 15 | FID Epoch 20 | FID Epoch 25 |
> |:-------------------------------- | -----------:| -----------:| -----------:| -----------:| -----------:| ------------:| ------------:| ------------:| ------------:|
> | CFM (EFM, M=1)                   |     378.00 |     181.25 |      67.88 |     29.44 |     15.30 |      4.20 |      3.08 |        2.51 |      2.28 |
> | EFM, M=128                       |      370.64 |     168.58 |     60.52 |     25.52 |     13.44 |      3.79 |      2.70 |      2.35 |      2.10 |
> | EFM, M=256                       |     370.94 |     169.71 |     61.88 |     25.73 |     13.48 |      3.73 |      2.76 |      2.33 |      2.08 |
> | EFM, M=1024 |     369.72 |     168.43 |     60.28 |     24.24 |     12.26 |      3.30 |      2.67 |      2.17 |      1.84 |
>
> We will add the results for FMNIST and tiny-imagenet by the end of the author/reviewer discussion period.
>
>
> ### Questions
> - *In L38, what is meant by the “deterministic regime” of the flow matching objective?*: We call the "deterministic regime" the times for which the average velocity field $\hat u$ only correlates with a single of the $n$ conditional fields $ u^\mathrm{cond}$, corresponding to the plot in Fig 1a, right.
> - Adding x-axis in Figure 3 : this has been done

---

> > ### Author Response · Authors · 2025-08-07
> > **Fashion MNIST dataset**
> >
> > We present the results for the Fashion MNIST dataset, and the conclusion mirrors that of the other dataset: **regressing against a more deterministic velocity field does not hinder generalization**.
> >
> > In this experiment, we used the same network architecture as in the MNIST experiment—specifically, the default U-Net with attention and timestep embedding from the torchcfm library. The Adam optimizer was used with all default parameters. A pretrained classifier, achieving 90% accuracy on Fashion MNIST, was employed to generate a lower-dimensional embedding of size 128. This embedding was then used to compute the FID between the test set and the generated set.
> >
> >
> >
> > | Method                           | FID Epoch 5 | FID Epoch 10 | FID Epoch 50 | FID Epoch 100 | FID Epoch 200 |
> > |:-------------------------------- | -----------:| ------------:| ------------:| -------------:| -------------:|
> > | CFM (EFM, M=1)                   |     253.56 |      48.67|      25.36 |       21.35 |       19.67 |
> > | EFM, M=128                       |     206.27 |      44.08 |      23.39 |       19.63 |       17.72 |
> > | EFM, M=256                       |     202.62 |      45.06 |      22.16 |       20.08 |       17.74 |
> > | EFM, M=512 |     194.66 |      44.19 |      22.10 |       18.93 |       16.85 |

---

> > > ### Author Response · Authors · 2025-08-07
> > > **AFHQ-cat dataset**
> > >
> > > While we initially considered reporting results on Tiny ImageNet, the dataset proved too computationally demanding to complete within the rebuttal period, requiring extensive tuning to obtain meaningful FID scores (we did not find any previous implementations for this specific dataset). As an alternative, we provide results on AFHQ-cat 64x64, as an additional baseline to support our findings.
> > >
> > >
> > >
> > > | Method     | FID Epoch 1656 | FID Epoch 2527 | FID Epoch 3312 |
> > > | ---------- | -------------- | -------------- | -------------- |
> > > | CFM (EFM, M=1)       | 68.20          | 66.85          | 64.91          |
> > > | OT-FM      | 70.07          | 67.60          | 67.12          |
> > > | EFM, M=128 | 67.12         | 65.38      | 64.28      |
> > > | EFM, M=256 | 67.07      | 66.58       | 64.55        |

---

> > > > ### Author Response · Authors · 2025-08-07
> > > >
> > > > As the author-reviewer discussion period is coming to a close, we’d be happy to clarify any remaining concerns.
> > > >
> > > > Thank you again for your time and feedback.

---

### Decision · Program_Chairs · 2025-09-17

**Decision:**

Accept (oral)

**Comment:**

(a) This paper investigates why flow-matching generative models generalize rather than memorizing training data. Two popular hypotheses are tested: 1) target stochasticity in the training loss drives generalization and 2) neural approximation mismatch with the closed-form velocity field induces smoother fields that interpolate between samples. Using the closed-form velocity field, the authors show that perfect fitting leads to memorization, whereas learned approximations deviate and yield better generalization. They introduce Empirical Flow Matching (EFM), a loss formulation that reduces stochasticity in training, and find that generalization still persists, thus refuting the stochasticity hypothesis. These claims are also verified experimentally.

(b) The method introduces a novel and insightful use of controlled loss variants to probe the role of stochasticity. The hypotheses are also validated by carefully designed and clearly presented experiments with convincing visualizations. The paper is well written. The proposed EFM loss provides a practically simpler alternative with stable training.

(c) The reviewers notice that the paper is mostly empirical in nature, as it does not provide a theoretical insight into why the mismatch between neural networks and the closed-form solution leads to generalization. The experiments could include a broader range of modalities, datasets and architectures. Some reviewers noted a lack of consistency in the narrative between rejecting and partially retaining the role of stochasticity.

(d) I recommend acceptance. All reviewers converged on positive assessments after rebuttal, with scores uniformly raised to 5. The paper provides meaningful contributions: it challenges a widely discussed explanation of generalization in flow-matching, proposes a simple alternative training loss, and supplies careful empirical evidence. While theoretical depth and broader validation are limited, the work clearly advances understanding in this area and is of interest to the generative modeling community.

(e) Across reviews, concerns centered on 1) the exact role of stochasticity in EFM, 2) missing computational cost evaluation, 3) limited dataset and modality scope, and 4) some presentation/narrative clarity issues. The authors clarified most of the important issues and the reviewers found these responses satisfactory. In fact, they raised their scores after rebuttal, with several stating that their primary concerns were resolved. The outstanding limitations (lack of broader experiments and deeper theoretical analysis) were acknowledged but not deemed critical for acceptance.